PLOS ·Medicine

# Cardiovascular–kidney–metabolic syndrome and all-cause and cardiovascular mortality: A retrospective cohort study

Min-Kuang Tsai[1,2], Juliana Tze-Wah Kao[1,2,3,4], Chung-Shun Wong[5,6], Chia-Te Liao[1,2,7], Wei-Cheng Lo[2,8,9], Kuo-Liong Chien[4,10,11], Chi-Pang Wen[12,13], Mai-Szu Wu[1,2,7]*, Mei-Yi Wu[1,2,7]*

**1** Division of Nephrology, Department of Internal Medicine, Shuang Ho Hospital, Taipei Medical University, New Taipei City, Taiwan, **2** Taipei Medical University Research Center of Urology and Kidney (TMU-RCUK), Taipei Medical University, Taipei, Taiwan, **3** Division of Nephrology, Department of Internal Medicine, Fu-Jen Catholic University Hospital, Fu-Jen Catholic University, New Taipei City, Taiwan, **4** Department of Internal Medicine, National Taiwan University Hospital and College of Medicine, Taipei, Taiwan, **5** Division of Emergency Medicine, Department of Emergency and Critical Care, Shuang Ho Hospital, Taipei Medical University, New Taipei City, Taiwan, **6** Department of Emergency Medicine, School of Medicine, College of Medicine, Taipei Medical University, Taipei, Taiwan, **7** Division of Nephrology, Department of Internal Medicine, School of Medicine, College of Medicine, Taipei Medical University, Taipei, Taiwan, **8** Master Program in Applied Epidemiology, College of Public Health, Taipei Medical University, Taipei, Taiwan, **9** School of Public Health, College of Public Health, Taipei Medical University, Taipei, Taiwan, **10** Institute of Epidemiology and Preventive Medicine, College of Public Health, National Taiwan University, Taipei, Taiwan, **11** Population Health Research Center, National Taiwan University, Taipei, Taiwan, **12** Institute of Population Health Sciences, National Health Research Institutes, Miaoli, Taiwan, **13** China Medical University Hospital, Taichung, Taiwan

* maiszuwu@gmail.com (M-SW); e220121@gmail.com (M-YW)

## Abstract

### Background

The American Heart Association recently issued guidelines introducing the concept of cardiovascular–kidney–metabolic (CKM) syndrome to emphasize the importance of multidisciplinary approaches to prevention, risk stratification, and treatment for these diseases. This study assessed the prevalence of CKM syndrome stages and the mortality risk associated with its components in a large Asian cohort.

### Methods and findings

We analyzed a retrospective cohort of 515,602 participants aged ≥20 years from a health screening program conducted between 1996 and 2017 in Taiwan. We assessed the associations of all-cause mortality, cardiovascular disease (CVD) mortality, and cause-specific mortality with CKM stages and its components—hypertension, diabetes mellitus, chronic kidney disease (CKD), metabolic syndrome, and hyperlipidemia. All participants were followed for a median of 16.5 years (interquartile range: 11.5, 21.2 years). Multivariate Cox proportional hazards models, adjusted for age, sex, educational level, smoking status, alcohol drinking status, and physical

**Data availability statement:** Researchers interested in obtaining the detail methodology codes and related data may contact Mr. Yuan Chieh Chuang at jackie_chuang@mjhrf.org. The data used in this research were authorized by MJ Health Research Foundation (Authorization Code: MJHRFB2014001C). The MJ Health Research Foundation administered the MJ Health Survey Database and MJ BioData, and the data are available at http://www.mjhrf.org/en/index.php?action=download&tid=3&cid=1.

**Funding:** MKT, the principal investigator, received research funding from the National Science and Technology Council (https://www.nstc.gov.tw/?l=en), under grant number NSTC 113-2314-B-038-024. MKT, the principal investigator, also received research funding from Taipei Medical University–Shuang Ho Hospital (https://shh.tmu.edu.tw/indexEng.aspx), under grant number 113FRP-20. This study is supported in part by the Taiwan Ministry of Health and Welfare Clinical Trial Center (MOHW110-TDU-B-212-124004). The funders had no role in study design, data collection and analysis, decision to publish, or preparation of the manuscript.

**Competing interests:** The authors have declared that no competing interests exist.

**Abbreviations:** ACR, albumin-to-creatinine ratio; AHA, American Heart Association; BMI, body mass index; CKD, chronic kidney disease; CKD-EPI, CKD Epidemiology Collaboration; CKM, cardiovascular–kidney–metabolic; CVD, cardio-vascular disease; eGFR, estimated glomerular filtration rate; NHANES, National Health and Nutrition Examination Survey; PAF, population-attributable fraction; PREVENT, Predicting Risk of cardiovascular disease EVENTs; RMST, restricted mean survival time; STROBE, Strengthening the Reporting of Observational Studies in Epidemiology.

activity groups, were used to calculate hazard ratios (HRs). We used Chiang's life table method to estimate years of life lost due to each CKM component. Among all participants, 257,535 (49.9%) were female. The majority of participants ($n = 368,578$ participants, (71.5%)) met criteria for CKM syndrome, with prevalence rates of 19.5%, 46.3%, 1.9%, and 3.8% for stages 1, 2, 3, and 4, respectively. CKM syndrome was associated with higher risks of all-cause mortality (HR: 1.33; 95% confidence interval, CI: 1.28, 1.39), CVD mortality (HR: 2.81; 95% CI: 2.45, 3.22), and incident end-stage kidney disease (ESKD) (HR: 10.15; 95% CI: 7.54, 13.67). Each additional CKM component was associated with a 22% increase in the risk of all-cause mortality (HR: 1.22; 95% CI: 1.21, 1.23), a 37% increase in the risk of CVD mortality (HR: 1.37; 95% CI: 1.35, 1.40) compared with those without any CKM components. In addition, each additional component reduced average life expectancy by 3 years. The population-attributable fractions of CKM syndrome were 18.7% (95% CI: 15.8, 21.7) for all-cause mortality and 55.0% (95% CI: 49.0, 60.4) for CVD mortality. We estimated that failing to include CKD in CKM syndrome could result in the missed attribution of 11% of CVD deaths. The primary limitation is that our analysis relied on baseline measurements only, without accounting for longitudinal changes.

## Conclusions

In the large cohort study, the prevalence of CKM syndrome and its components were associated with risks of all-cause mortality, CVD mortality, and ESKD. These findings highlight the clinical need for integrated care within CKM health.

## Author summary

### Why was this study done?

- In November 2023, the American Heart Association (AHA) introduced new guidelines emphasizing the importance of cardiovascular–kidney–metabolic (CKM) health.

- As CKM syndrome is a newly defined concept, understanding its impact on mortality risk and cardiovascular disease (CVD) outcomes, as well as the distribution of CKM stages in the general population is crucial.

### What did the researchers do and find?

- We investigated the epidemiological distribution of CKM syndrome and its associations with mortality outcomes in a large Asian cohort.

- We assessed mortality risk on the basis of the number of CKM components (hypertension, chronic kidney disease, diabetes, metabolic syndrome, and hyperlipidemia) present. CKM syndrome was prevalent in 71.5% of participants,

with nearly 90% of those aged ≥55 years affected. Among all participants, one-fifth (22.7%) of the participants had at least two components, while one-ninth (11.7%) had at least three.

- We found that the risks of all-cause mortality, CVD mortality, and incident end-stage kidney disease (ESKD) increased with the number of CKM components present.

## What do these findings mean?

- CKM syndrome should not be treated as a set of isolated conditions. Physicians should adopt an interdisciplinary approach, addressing all five components when any single disease component is diagnosed to ensure comprehensive, cross-specialty care.

- The primary limitation of this study is that our analysis relied on baseline measurements only, without accounting for longitudinal changes. Future research should explore how CKM stage transitions over time influence health outcomes for a more comprehensive understanding of its implications.

## Introduction

Global mortality from cardiovascular diseases (CVDs) has been on the rise, increasing from 12.4 million in 1990 to 19.8 million in 2022 [1]. Population aging and an increased prevalence of chronic diseases have exacerbated the burden of CVDs [2]. Approximately 850 million individuals worldwide have chronic kidney disease (CKD) [3]. Among individuals with CKD, 2.6 million require dialysis, which is given to treat end-stage kidney disease (ESKD) [4]. In addition, approximately 537 million individuals worldwide have diabetes [5,6]. More than 1.56 billion individuals worldwide have hypertension, which is a major risk factor for global mortality from CVD [7].

In November 2023, the American Heart Association (AHA) published new guidelines that emphasize the importance of cardiovascular–kidney–metabolic (CKM) health [8,9]. Patients with CVD often have comorbid hypertension, kidney disease, and metabolic disorders, which share common underlying mechanisms and have a critical impact on CVD incidence and mortality [10–13]. This CKM syndrome is defined and staged based on the presence of clinical characteristics (e.g., obesity) and comorbidities (e.g., hypertension, diabetes, and CKD). Because CKM syndrome is a newly defined concept, it is essential to understand its impact on mortality risk and CVD outcomes, as well as the distribution of different CKM stages in the general population. Additionally, CKM introduces an interdisciplinary approach to health, shifting the focus from individual disease factors to the interactions among multiple diseases. CKM syndrome emphasize the importance of multidisciplinary approaches to prevention, risk stratification, and treatment for these diseases. Unlike previous models for CVD risk prediction, CKM syndrome incorporates kidney disease-related risk factors. Recently, international scientific societies have advocated for the inclusion of kidney disease in the World Health Organization's (WHO) framework on major non-communicable disease determinants of premature mortality [14]. However, while incorporating CKD into this framework acknowledges its significance, concerns remain regarding its predictive utility for all-cause and CVD outcomes, particularly given the limited public awareness of CKD [15]. Recent therapeutic advances further support the rationale for an integrated CKM framework, as several new medications demonstrate beneficial effects across multiple disease domains [16]. Sodium-glucose cotransporter-2 inhibitors, for example, have shown remarkable efficacy in reducing cardiovascular events, slowing CKD progression, and improving glycemic control [17]. Similarly, glucagon-like peptide-1 receptor agonists [18] and nonsteroidal mineralocorticoid receptor antagonists exhibit protective effects across multiple organ systems [19]. These findings suggest that optimal treatment strategies should consider the entire CKM spectrum rather than focusing on isolated conditions.

Therefore, this study investigates the combinations of different CKM components and their association with mortality risk, including hypertension, CKD, diabetes mellitus, metabolic syndrome, and hyperlipidemia. Previously, we used a large

cohort to examine the prevalence and mortality burden of CKD [20]. We also assessed awareness of each CKM component. In addition to evaluating awareness of diabetes and hypertension, we also examine CKD awareness in this study, particularly given that public awareness of CKD is generally lower compared to other CKM components [20].

## Methods

In a retrospective cohort of 515,602 participants aged ≥20 years, medical screenings were conducted from 1996 to 2017, resulting in 8.30 million person-years of follow-up data on all-cause mortality, CVD, and ESKD over a 25-year period. The participants underwent standardized physical examinations and biochemical tests and completed questionnaires on lifestyle habits and medical history (S1 Fig).

The study cohort was derived from a private health screening program in Taiwan. Standardized screening procedures were implemented using identical instruments, ensuring uniformity in data collection [20,21]. All results were centrally managed and securely stored. Participants were non-institutionalized, generally healthy individuals from diverse backgrounds. They were encouraged to undergo annual health screenings, during which identical questionnaires were administered at each visit. However, for this study, only baseline data from the initial visit were used for analysis. This study was approved by the Institutional Review Board of Taipei Medical University (Approval Number: N202404037). All participants gave written consent to analysis their unidentifiable data before being included in this study. The details of the cohort have been previously reported [21,22]. All personally identifiable information was removed, and data remained anonymized throughout the study process. This study is reported as per the Strengthening the Reporting of Observational Studies in Epidemiology (STROBE) guideline (S1 File).

The definition of CKM health, based on the guidelines published by the AHA in November 2023, includes specific criteria for individuals of Asian ethnicity regarding body mass index (BMI) and waist circumference [8,9]. The stages of CKM are defined as follows (Table 1). Stage 1: BMI ≥ 23 kg/m$^2$, waist circumference ≥80 cm for female and ≥90 cm for male, or fasting blood glucose level = 100–124 mg/dL. Stage 2: presence of hypertension, diabetes, metabolic syndrome, high triglyceride levels (≥135 mg/dL), or any stage of CKD. Stage 3: with a Predicting Risk of cardiovascular disease EVENTs (PREVENT) score indicating a 10-year risk of CVD exceeding 20% [23,24] or has a kidney function with an estimated glomerular filtration rate (eGFR) of ≤30 mL/min/1.73 m$^2$. Stage 4: self-reports medication use or seeks medical attention for coronary heart disease, heart failure, or stroke or has CKD with an eGFR of ≤30 mL/min/1.73 m$^2$. Because our cohort is relatively young, the generalizability of the prevalence estimates may be affected. To account for this, we have included age- and education-adjusted prevalence estimates to better project the prevalence in this population.

Screened hypertension or under antihypertensive therapy was defined as a history of hypertension, the use of anti-hypertensive medication, or a systolic blood pressure of ≥130 mmHg or a diastolic blood pressure of ≥80 mmHg. CKD was determined on the basis of eGFR by using the CKD Epidemiology Collaboration (CKD-EPI) formula [25], and proteinuria levels were detected using a dipstick test, categorized as negative, trace, 1+, 2+, or 3+ [20]. Screened diabetes or under antidiabetic therapy was defined as a history of diabetes, the use of diabetes medication, or a fasting blood glucose level of ≥126 mg/dL. Screened hyperlipidemia or under hyperlipidemia therapy was defined as a history of hyperlipidemia, the use of lipid-lowering medication, or a triglyceride level of ≥135 mg/dL. According to the National Cholesterol Education Program, Adult Treatment Panel III guidelines [26], metabolic syndrome was defined as the presence of at least three of the following criteria: abdominal obesity, hypertension, elevated triglycerides, low concentration of high-density lipoprotein cholesterol, and elevated fasting blood glucose. Awareness of the chronic conditions was defined as those who had each of the CKM components who answered yes to this question. In this study, the definition of CKM is based on the AHA's criteria, but there are several key differences to note: The cut-off points for BMI and waist circumference in this study follow the AHA-defined Asian criteria. The definition of fasting blood glucose for CKM Stage 1 (100–124 mg/dL) is consistent with the AHA definition. However, since HbA1c measurements are not available in this cohort, we did not include the criterion of HbA1c between 5.7% and 6.4%. The definition of CKD

**Table 1. Definition of CKM syndrome stages in this study.**

| CKM syndrome stages | Definition |
| --- | --- |
| Stage 0: No CKM risk factors | Individuals with normal BMI and waist circumference, normoglycemia, normotension, a normal lipid profile, and no evidence of CKD or subclinical or clinical CVD |
| Stage 1: Excess or dysfunctional adiposity | Individuals with overweight/obesity, abdominal obesity, or dysfunctional adipose tissue, without the presence of other metabolic risk factors or CKD. BMI[a] ≥ 23 kg/m², waist circumference[a] ≥80 cm for female and ≥90 cm for male, or fasting blood glucose level = 100–124 mg/dL.[b] |
| Stage 2: Metabolic risk factors and CKD | Presence of hypertension,[c] diabetes,[d] metabolic syndrome,[e] high triglyceride levels (≥135 mg/dL),[f] or any stage of CKD[g] |
| Stage 3: Subclinical CVD in CKM | With a Predicting Risk of cardiovascular disease EVENTs (PREVENT) score indicating a 10-year risk of CVD exceeding 20% or has a kidney function with an eGFR of ≤30 mL/min/1.73 m². |
| Stage 4: Clinical CVD in CKM | Self-reports medication use or seeks medical attention for coronary heart disease, heart failure, or stroke or has CKD with an eGFR of ≤30 mL/min/1.73 m². |

In this study, the definition of CKM is based on the American Heart Association's criteria, but there are several key differences to note:

- The cut-off points for BMI and waist circumference in this study follow the AHA-defined *Asian criteria*.

- The definition of fasting blood glucose for CKM Stage 1 (100–124 mg/dL) is consistent with the AHA definition. However, since HbA1c measurements are not available in this cohort, we did not include the criterion of HbA1c between 5.7% and 6.4%.

- The definition of CKD in CKM Stages 2–4 is based on the CKD Epidemiology Collaboration (CKD-EPI) formula, which aligns with the AHA definition. However, for proteinuria assessment, the original AHA definition uses albumin-to-creatinine ratio (ACR), whereas this study relies on dipstick testing.

- The definition of Clinical CVD in CKM Stage 4 in this study is based on self-reported medication use or seeking medical attention for coronary heart disease, heart failure, or stroke. This differs from the AHA definition, which requires a clinical diagnosis by physicians.

[a]Asian criteria.

[b]HbA1c are not available in this cohort.

[c]Hypertension was defined as a history of hypertension, the use of antihypertensive medication, or a systolic blood pressure of ≥130 mmHg or a diastolic blood pressure of ≥80 mmHg.

[d]Diabetes was defined as a history of diabetes, the use of diabetes medication, or a fasting blood glucose level of ≥126 mg/dL.

[e]Metabolic syndrome was defined as the presence of at least three of the following criteria: waist circumference ≥80 cm for female and ≥90 cm for male; systolic blood pressure ≥130 mmHg or diastolic blood pressure ≥80 mmHg and/or use of antihypertensive medications; fasting blood glucose ≥100 mg/dL; HDL cholesterol <40 mg/dL for male and <50 mg/dL for female; triglycerides ≥150 mg/dL.

[f]High triglyceride levels was defined as a history of hyperlipidemia, the use of lipid-lowering medication, or a triglyceride level of ≥135 mg/dL.

[g]CKD was determined on the basis of eGFR by using the CKD Epidemiology Collaboration (CKD-EPI) formula, and proteinuria levels were detected using a dipstick test, categorized as negative, trace, 1+, 2+, or 3+.

CKM: cardiovascular–kidney–metabolic syndrome; BMI: body mass index; CKD: chronic kidney disease; CVD: cardiovascular disease; PREVENT: predicting Risk of cardiovascular disease EVENTs; eGFR: estimated glomerular filtration rate.

in CKM Stages 2–4 is based on the CKD-EPI formula, which aligns with the AHA definition. However, for proteinuria assessment, the original AHA definition uses albumin-to-creatinine ratio (ACR), whereas this study relies on dipstick testing. The definition of Clinical CVD in CKM Stage 4 in this study is based on self-reported medication use or seeking medical attention for coronary heart disease, heart failure, or stroke. This differs from the AHA definition, which requires a clinical diagnosis by physicians.

## Dependent variables

Taiwan maintains a death registry on the basis of citizen death certificates. We linked the unique identification number to this database to obtain mortality outcomes. Causes of death in Taiwan are categorized using *International Classification of Diseases, Ninth Revision* or *Tenth Revision* codes. Causes of death were classified according to ICD-9 and ICD-10 codes and included all-cause mortality (ICD-9: 001–998; ICD-10: A00–Y98), CVD (390–459; I00–I99), heart diseases (410–414; I20-I25), stroke (430–438; I60–I69) and expanded cardiovascular diseases, which encompassed cardiovascular diseases

plus diabetes (ICD-9: 250) and kidney diseases (ICD-9: 580–589). We included individuals diagnosed with ESKD as well as those with a kidney transplant.

## Covariates

The participants completed self-administered questionnaires, underwent physical examinations, and provided blood and urine samples. They were categorized on the basis of educational level into four groups: middle school or below, high school, junior college, and college or graduate school. Smoking status was classified as nonsmoker, ex-smoker, or current smoker, and alcohol consumption was classified as nondrinker, occasional drinker, or regular drinker. Data on physical activity levels were collected by using questions on the weekly duration and types of activity. The participants were categorized into five activity levels on the basis of their weekly MET-hour totals: inactive, low, medium, high, and very high [27].

## Statistical analysis

The entry time was considered as the date of recruitment, and the exit time was the end of follow-up (December 31, 2020) or the date of death, whichever came earlier. For participants with ESKD, the date of starting dialysis or receiving a kidney transplant was used as the exit date. Those with history of kidney disease and undergoing dialysis before the screening were excluded for the ESKD outcome. We performed multivariate analysis by using the Cox proportional hazards model to calculate hazard ratios (HRs) for all-cause, CVD (including heart disease and stroke), diabetes, and kidney disease mortality and incident ESKD. All Cox proportional hazards models were adjusted for age, sex, educational level, smoking status, drinking status, and physical activity groups. In addition, we examined the proportional hazards assumption by plotting Kaplan–Meier survival curves and testing for interactions between CKM stages and follow-up time (S2 and S3 Figs). We also performed restricted mean survival time (RMST) regression model for each CKM stage. To assess the risks of different CKM components, including hypertension, CKD, diabetes, metabolic syndrome, and hyperlipidemia, we included participants without any of these conditions into the reference group. Life expectancy is calculated using life tables, which estimate the expected remaining years of life for individuals at a given age based on mortality rates. We estimated age-specific mortality rates based on different CKM stages and sex-specific stratifications to determine the probability of survival. Chiang's life table method was employed to calculate years of life lost for each CKM stages and components (S2 File). In this study, life expectancy refers to the average expected remaining years of life for individuals at ages 30, 40, 50, 60, and 70 years. Previously, we used our cohort data to estimate life expectancy based on at least 30 different health risk factors [28,29]. The population-attributable fraction (PAF) of CKM was defined as $P \times (HR - 1)/(1 + P \times [HR - 1])$ where $P$ is the prevalence of CKM components and HR is the HR. The total PAF of CKM was estimated by the following formula:

$$\frac{\sum_{i=1}^{n} Pi(RRi - 1)}{\sum_{i=1}^{n} Pi(RRi - 1) + 1}$$

where $i$ signifies the stages of CKM ($i = 1, \ldots, n$) [30].

We conducted an analysis among participants who had undergone two health examinations. Spearman's rank correlation coefficient was calculated between the CKM stages at baseline and those at the second examination. We have incorporated a time-dependent analysis for participants with a second examination as a sensitivity analysis. This approach accounts for changes in CKM stages over time and the duration of follow-up, allowing us to further evaluate the association between CKM stages and mortality outcomes. All statistical analyses were performed using SAS 9.4. A two-sided $p$-value of 0.05 indicated statistical significance.

## Results

The study cohort included 515,602 participants, with a mean (standard deviation) age of 40.3 (13.4) years at recruitment. Among them, 257,535 (49.9%) were female. The majority of participants (n = 368,578 participants, (71.5%)) met criteria for CKM syndrome, with 19.5%, 46.3%, 1.9%, and 3.8% classified as CKM stages 1, 2, 3, and 4, respectively. Distribution of sociodemographic characteristics stratified by CKM stages were shown in Table 2. The prevalence of CKM syndrome increased with age, exceeding 90% among participants aged 55 years or older, with CKM stages 3–4 becoming more prevalent in this age group (Figs 1 and S4). Participants with CKM syndrome, compared to those without the syndrome, were older (aged 60 or above: 15.0% versus 1.7%), more likely to be male (58.7% versus 28.3%), have a middle school education or lower (27.5% versus 7.8%), be current smokers (25.2% versus 17.1%), and consume alcohol regularly (10.6% versus 4.7%). Moreover, the prevalence of physical inactivity was lower among participants with CKM syndrome (48.9%) than among those without CKM (55.9%).

Table 3 presents the individual causes of death and the risk of ESKD across different stages of CKM. A total of 41,589 all-cause deaths, 8,825 CVD deaths, and 3,334 incidences of ESKD were identified over a median follow-up period of 16.5 years (interquartile range, IQR: 11.5, 21.2). The risk of all-cause mortality was higher among participants with CKM syndrome than those with CKM stage 0 (HR: 1.33; 95% CI: 1.28, 1.39). Furthermore, the risks of CVD mortality and ESKD were higher among participants with CKM than those with CKM stage 0 (CVD mortality HR: 2.81; 95% CI: 2.45, 3.22 and ESKD HR: 10.15; 95% CI: 7.54, 13.67). Compared to participants with CKM Stage 0, the risk of death for CKM stages 1–5 increased with the severity of the stage. The risks of diabetes and expanded CVD were significantly higher in participants with CKM stage 1 than in those with CKM stage 0 (HR: 3.60; 95% CI: 1.66, 7.84 for diabetes mortality and HR: 1.21; 95% CI: 1.03, 1.43 for expanded CVD mortality). However, there was no significant difference in the risk of all-cause (HR: 0.96; 95% CI: 0.91, 1.02) and CVD death (HR: 1.13; 95% CI: 0.95, 1.35) between CKM Stage 1 and Stage 0 participants. Mortality risk was significantly higher in participants with CKM stage 2 than in those with CKM stage 0 (all-cause mortality HR: 1.36; 95% CI: 1.30, 1.42; CVD mortality HR: 2.89; 95% CI: 2.51, 3.32; and ESKD HR: 12.34; 95% CI: 9.14, 16.67). Across these different causes of death, the risk for CKM Stages 3–4 was higher than for CKM Stage 2 (S5 Fig).

To further assess survival differences (S1 Table), we analyzed RMST. CKM Stage 0 serves as the reference, with an RMST of 23.395 years. CKM stage 1 showed a slight reduction in RMST (23.248 years; difference: −0.147, 95% CI: −0.19, −0.11), while CKM Stage 2 had a more pronounced decline (23.140 years; difference: −0.255, 95% CI: −0.29, −0.22). The reduction was more substantial in CKM stage 3 (21.161 years; difference: −2.234, 95% CI: −2.44, −2.03) and stage 4 (19.043 years; difference: −4.352, 95% CI: −4.72, −3.98). These results indicate a progressive decline in survival as CKM stage advances. The robustness of the findings was assessed using multiple imputed data to account for missing baseline information and outcomes (S2 Table). The results for those with multiple imputation and those excluded due to missing data were generally similar. Additionally, we conducted a sensitivity analysis considering the use of analgesics (2.1%) and cancer history (1.1%). After incorporating the variables, the main results, including all-cause mortality and cardiovascular mortality, remained unchanged (S3 Table).

For all-cause mortality, compared with the absence of all the five CKM components, the presence of each additional CKM component was associated with a 22% increase in the risk of death (HR: 1.22; 95% CI: 1.21, 1.23) and a 37% increase in the risk of CVD mortality (HR: 1.37; 95% CI: 1.35, 1.40). The increase in risk was even more pronounced for diabetes mortality, kidney-related mortality, and ESKD incidence. The aforementioned risk increased exponentially (Table 4 and Fig 2). We also examined the risk of each individual CKM components (panels 1–2 of S4 Table). Participants with diabetes alone had the highest risk of all-cause and CVD mortality (HR: 1.86; 95% CI: 1.67, 2.06 for all-cause and HR: 2.42; 95% CI: 1.84, 3.18). Participants with hypertension alone had the risk of CVD mortality (HR: 2.40; 95% CI: 2.18, 2.65). Participants with CKD or diabetes had the highest risk of ESKD (HR: 16.35; 95% CI: 12.51, 21.36 among participants with CKD and HR: 43.77; 95% CI: 31.64, 60.56 among participants with diabetes).

**Table 2. Characteristics of participants stratified by cardiovascular–kidney–metabolic syndrome stage.**

| | | Non-CKM | | Total CKM | | Stage 1 | | Stage 2 | | Stage 3 | | Stage 4 | |
|---|---|---|---|---|---|---|---|---|---|---|---|---|---|
| | | 147,024 | (28.5) | 368,578 | (71.5) | 100,585 | (19.5) | 238,647 | (46.3) | 9,925 | (1.9) | 19,421 | (3.8) |
| | | *N, (%)* | | | | | | | | | | | |
| Age (years) | 20–39 | 119,417 | (81.2) | **176,944** | **(48.0)** | 64,822 | (64.4) | 108,680 | (45.5) | 41 | (0.4) | 3,401 | (17.5) |
| | 40–59 | 25,142 | (17.1) | **136,269** | **(37.0)** | 31,228 | (31.0) | 97,247 | (40.7) | 1,010 | (10.2) | 6,784 | (34.9) |
| | 60 or more | 2,465 | (1.7) | **55,365** | **(15.0)** | 4,535 | (4.5) | 32,720 | (13.7) | 8,874 | (89.4) | 9,236 | (47.6) |
| Sex | Male | 41,626 | (28.3) | **216,441** | **(58.7)** | 52,856 | (52.5) | 147,330 | (61.7) | 6,941 | (69.9) | 9,314 | (48.0) |
| | Female | 105,398 | (71.7) | **152,137** | **(41.3)** | 47,729 | (47.5) | 91,317 | (38.3) | 2,984 | (30.1) | 10,107 | (52.0) |
| Education | Middle school or below | 10,654 | (7.8) | **94,628** | **(27.5)** | 14,743 | (15.8) | 63,044 | (28.4) | 6,700 | (69.2) | 10,141 | (53.5) |
| | High school | 30,444 | (22.2) | **72,567** | **(21.1)** | 19,873 | (21.3) | 47,834 | (21.5) | 1,468 | (15.2) | 3,392 | (17.9) |
| | Junior college | 33,592 | (24.5) | **63,430** | **(18.4)** | 19,091 | (20.4) | 41,450 | (18.7) | 778 | (8.0) | 2,111 | (11.1) |
| | College or above | 62,354 | (45.5) | **113,766** | **(33.0)** | 39,812 | (42.6) | 69,913 | (31.5) | 738 | (7.6) | 3,303 | (17.4) |
| Smoking status | Non-smoker | 106,886 | (79.2) | **227,464** | **(67.3)** | 66,487 | (72.3) | 143,147 | (65.8) | 5,043 | (50.8) | 12,787 | (69.4) |
| | Ex-smoker | 4,923 | (3.6) | **25,350** | **(7.5)** | 5,499 | (6.0) | 16,359 | (7.5) | 1,314 | (13.2) | 2,178 | (11.8) |
| | Current smoker | 23,093 | (17.1) | **84,971** | **(25.2)** | 20,012 | (21.8) | 57,943 | (26.6) | 3,568 | (35.9) | 3,448 | (18.7) |
| Drinking status | Non-drinker | 115,823 | (87.2) | **253,780** | **(76.8)** | 73,740 | (81.8) | 159,996 | (75.1) | 6,431 | (68.1) | 13,613 | (76.0) |
| | Occasional drinker | 10,765 | (8.1) | **41,787** | **(12.6)** | 10,419 | (11.6) | 28,645 | (13.5) | 1,032 | (10.9) | 1,691 | (9.4) |
| | Regular drinker | 6,262 | (4.7) | **34,867** | **(10.6)** | 6,003 | (6.7) | 24,278 | (11.4) | 1,976 | (20.9) | 2,610 | (14.6) |
| Physical activity | Inactive | 78,180 | (55.9) | **170,635** | **(48.9)** | 46,831 | (49.4) | 111,707 | (49.5) | 3,830 | (39.9) | 8,267 | (44.1) |
| | Low | 37,765 | (27.0) | **86,197** | **(24.7)** | 25,289 | (26.7) | 55,203 | (24.4) | 1,713 | (17.9) | 3,992 | (21.3) |
| | Medium | 16,076 | (11.5) | **55,705** | **(16.0)** | 14,222 | (15.0) | 35,540 | (15.7) | 2,207 | (23.0) | 3,736 | (19.9) |
| | High | 4,870 | (3.5) | **22,822** | **(6.5)** | 5,002 | (5.3) | 14,642 | (6.5) | 1,303 | (13.6) | 1,875 | (10.0) |
| | Very high | 2,932 | (2.1) | **13,626** | **(3.9)** | 3,416 | (3.6) | 8,800 | (3.9) | 543 | (5.7) | 867 | (4.6) |
| Body mass index (kg/m$^2$) | <18.5 | 29,718 | (20.2) | **13,202** | **(3.6)** | 3,161 | (3.1) | 8,765 | (3.7) | 339 | (3.4) | 937 | (4.8) |
| | 18.5–22.9 | 117,306 | (79.8) | **107,946** | **(29.3)** | 24,437 | (24.3) | 74,738 | (31.3) | 2,885 | (29.1) | 5,886 | (30.3) |
| | 23–29.9 | 0 | (0.0) | **224,796** | **(61.0)** | 70,043 | (69.6) | 137,639 | (57.7) | 6,066 | (61.1) | 11,048 | (56.9) |
| | ≥30 | 0 | (0.0) | **22,634** | **(6.1)** | 2,944 | (2.9) | 17,505 | (7.3) | 635 | (6.4) | 1,550 | (8.0) |
| Waist (cm) | Male < 90 cm; Female < 80 cm | 147,024 | (100.0) | **292,235** | **(79.3)** | 88,046 | (87.5) | 184,590 | (77.3) | 6,262 | (63.1) | 13,337 | (68.7) |
| | Male ≥90 cm; Female ≥80 cm | 0 | (0.0) | **76,343** | **(20.7)** | 12,539 | (12.5) | 54,057 | (22.7) | 3,663 | (36.9) | 6,084 | (31.3) |
| Screened hypertension or under antihypertensive therapy* | None | 147,024 | (100.0) | **182,697** | **(49.6)** | 100,585 | (100.0) | 75,183 | (31.5) | 916 | (9.2) | 6,013 | (31.0) |
| | Yes | 0 | (0.0) | **185,881** | **(50.4)** | 0 | (0.0) | 163,464 | (68.5) | 9,009 | (90.8) | 13,408 | (69.0) |
| CKD | Negative | 147,024 | (100.0) | **318,941** | **(86.5)** | 100,585 | (100.0) | 200,752 | (84.1) | 4,297 | (43.3) | 13,307 | (68.5) |
| | Stage 1 | 0 | (0.0) | **11,713** | **(3.2)** | 0 | (0.0) | 11,288 | (4.7) | 104 | (1.0) | 321 | (1.7) |

*(Continued)*

**Table 2.** (Continued)

| | | Cardiovascular-kidney-metabolic syndrome | | | | | | | | | | |
|---|---|---|---|---|---|---|---|---|---|---|---|---|
| | | **Non-CKM** | | **Total CKM** | | **Stage 1** | | **Stage 2** | | **Stage 3** | | **Stage 4** |
| | | **147,024** | **(28.5)** | **368,578** | **(71.5)** | **100,585** | **(19.5)** | **238,647** | **(46.3)** | **9,925** | **(1.9)** | **19,421** | **(3.8)** |
| | Stage 2 | 0 | (0.0) | **16,537** | **(4.5)** | 0 | (0.0) | 14,495 | (6.1) | 887 | (8.9) | 1,155 | (5.9) |
| | Stage 3 | 0 | (0.0) | **20,388** | **(5.5)** | 0 | (0.0) | 12,112 | (5.1) | 4,637 | (46.7) | 3,639 | (18.7) |
| | Stage 4 | 0 | (0.0) | **701** | **(0.2)** | 0 | (0.0) | 0 | (0.0) | 0 | (0.0) | 701 | (3.6) |
| | Stage 5 | 0 | (0.0) | **298** | **(0.1)** | 0 | (0.0) | 0 | (0.0) | 0 | (0.0) | 298 | (1.5) |
| Screened diabetes or under antidiabetic therapy* | None | 147,024 | (100.0) | **341,899** | **(92.8)** | 100,585 | (100.0) | 220,054 | (92.2) | 5,532 | (55.7) | 15,728 | (81.0) |
| | Yes | 0 | (0.0) | **26,679** | **(7.2)** | 0 | (0.0) | 18,593 | (7.8) | 4,393 | (44.3) | 3,693 | (19.0) |
| Metabolic syndrome | None | 147,024 | (100.0) | **297,671** | **(80.8)** | 100,585 | (100.0) | 180,962 | (75.8) | 4,228 | (42.6) | 11,896 | (61.3) |
| | Yes | 0 | (0.0) | **70,907** | **(19.2)** | 0 | (0.0) | 57,685 | (24.2) | 5,697 | (57.4) | 7,525 | (38.7) |
| Screened hyperlipidemia or under hyperlipidemia therapy* | None | 147,024 | (100.0) | **234,832** | **(63.7)** | 100,585 | (100.0) | 117,776 | (49.4) | 5,073 | (51.1) | 11,398 | (58.7) |
| | Yes | 0 | (0.0) | **133,746** | **(36.3)** | 0 | (0.0) | 120,871 | (50.6) | 4,852 | (48.9) | 8,023 | (41.3) |
| | Mean (SD) | | | | | | | | | | | |
| Age (years) | | 33.1 | (8.9) | 43.1 | (13.9) | 37.6 | (10.8) | 43.3 | (13.1) | 70.1 | (8.4) | 55.9 | (15.0) |
| BMI (kg/m²) | | 20.0 | (1.7) | 24.4 | (3.5) | 24.1 | (2.9) | 24.5 | (3.7) | 24.6 | (3.5) | 24.5 | (3.9) |
| Waist circumference (cm) | | 68.1 | (5.9) | 80.8 | (9.9) | 78.5 | (8.3) | 81.5 | (10.2) | 86.3 | (9.5) | 82.3 | (11.0) |
| WHR | | 0.75 | (0.1) | 0.84 | (0.1) | 0.81 | (0.1) | 0.84 | (0.1) | 0.91 | (0.1) | 0.86 | (0.1) |
| SBP (mmHg) | | 106.6 | (10.4) | 124.8 | (18.8) | 111.6 | (10.0) | 128.5 | (17.5) | 152.5 | (24.5) | 133.4 | (23.1) |
| DBP (mmHg) | | 64.5 | (7.2) | 75.5 | (11.5) | 67.2 | (6.8) | 78.6 | (11.1) | 82.4 | (14.3) | 77.2 | (12.8) |
| Fasting glucose (mg/dL) | | 90.5 | (5.4) | 103.3 | (25.8) | 98.2 | (7.7) | 103.9 | (27.7) | 128.2 | (49.3) | 109.8 | (34.1) |
| eGFR (mL/min/1.73 m²) | | 97.0 | (14.9) | 86.2 | (16.9) | 91.6 | (14.6) | 85.9 | (15.9) | 62.2 | (15.0) | 73.5 | (22.3) |
| Total cholesterol (mg/dL) | | 180.3 | (31.6) | 199.3 | (37.8) | 188.1 | (32.7) | 203.6 | (38.5) | 204.9 | (41.9) | 202.8 | (41.1) |
| Triglyceride (mg/dL) | | 68.8 | (23.5) | 135.5 | (113.7) | 83.2 | (25.8) | 155.5 | (126.6) | 157.3 | (134.0) | 148.3 | (126.8) |
| LDL (mg/dL) | | 105.8 | (28.8) | 123.1 | (33.6) | 117.6 | (30.3) | 125.0 | (34.3) | 128.4 | (36.4) | 124.0 | (35.6) |
| HDL (mg/dL) | | 60.7 | (15.5) | 51.3 | (14.6) | 54.5 | (14.2) | 50.3 | (14.5) | 45.9 | (13.3) | 51.1 | (15.3) |
| Resting heart rate (beats/min) | | 70.6 | (9.8) | 72.6 | (11.0) | 70.1 | (9.9) | 73.5 | (11.1) | 74.1 | (12.7) | 72.9 | (12.1) |

*Screened hypertension or under antihypertensive therapy was defined as a history of hypertension, the use of antihypertensive medication, or a systolic blood pressure of ≥130 mmHg or a diastolic blood pressure of ≥80 mmHg. Screened diabetes or under antidiabetic therapy was defined as a history of diabetes, the use of diabetes medication, or a fasting blood glucose level of ≥126 mg/dL. Screened hyperlipidemia or under hyperlipidemia therapy was defined as a history of hyperlipidemia, the use of lipid-lowering medication, or a triglyceride level of ≥135 mg/dL.

CKM: cardiovascular–kidney–metabolic syndrome; N: number of participants; CKD: chronic kidney disease; SD: standard deviation; BMI: body mass index; WHR: waist to hip ratio; SBP: systolic blood pressure; DBP: diastolic blood pressure; eGFR: estimated glomerular filtration rate; LDL: low-density lipoprotein cholesterol; HDL: high-density lipoprotein cholesterol.

We evaluated the differential mortality risk associated with CKM by comparing the remaining life expectancy across different CKM stages and components separately for male and female participants (S5 Table). Among males starting at age 30, those with CKM stages 1, 2, 3, and 4 had, respectively, 0.2 (95% CI: −0.03 to 0.39), 2.5 (2.36 to 2.55), 6.2 (5.03 to 7.38), and 7.0 (6.42 to 7.58) fewer years of life expectancy compared with those at CKM stage 0. For females at the

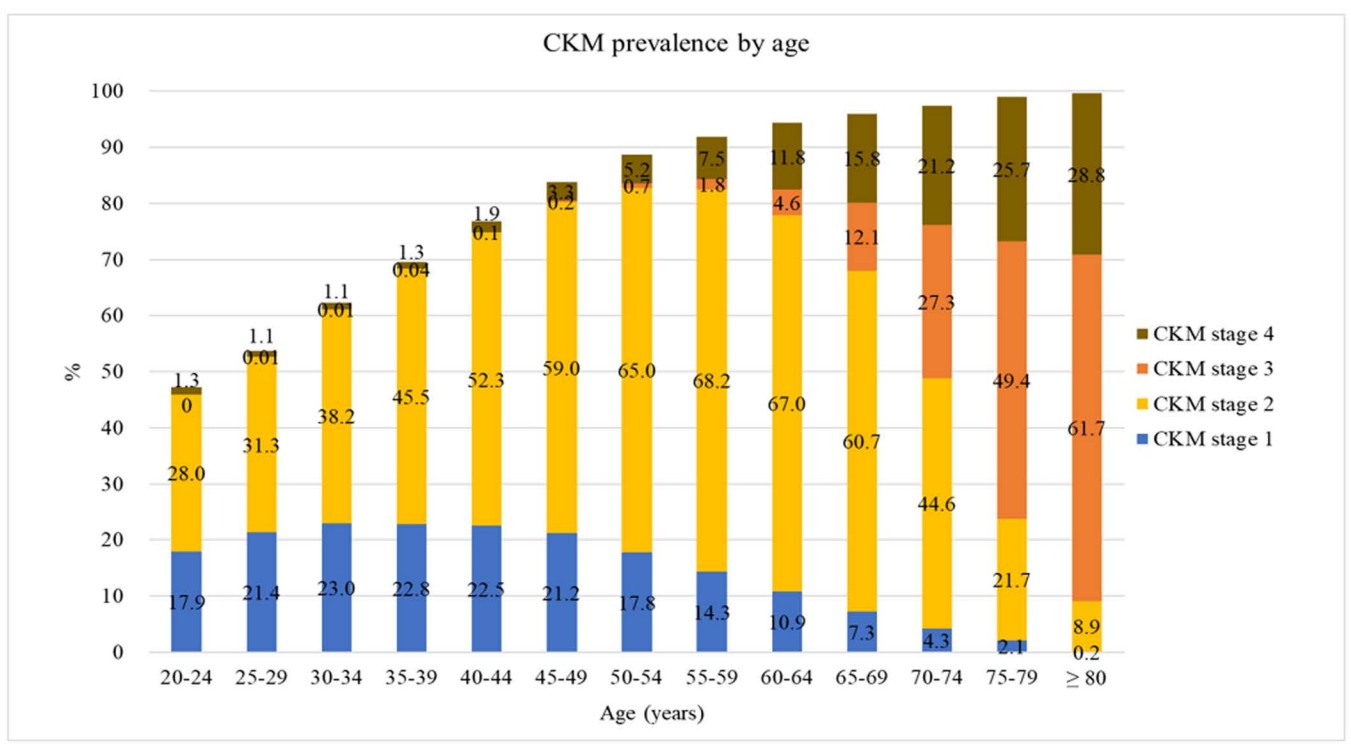

**Fig 1. Age-stratified prevalence of cardiovascular–kidney–metabolic syndrome.**

| Age (years) | N of participants | CKM Prevalence | | CKM stage 1 | | CKM stage 2 | | CKM stage 3 | | CKM stage 4 | |
|---|---|---|---|---|---|---|---|---|---|---|---|
| | | n of CKM | (%) | n of CKM | (%) | n of CKM | (%) | n of CKM | (%) | n of CKM | (%) |
| 20-24 | 33,915 | 16,030 | (47.3) | 6,082 | (17.9) | 9,511 | (28.0) | 0 | (0.0) | 437 | (1.3) |
| 25-29 | 91,747 | 49,314 | (53.7) | 19,641 | (21.4) | 28,690 | (31.3) | 5 | (0.0) | 978 | (1.1) |
| 30-34 | 98,081 | 61,056 | (62.3) | 22,559 | (23.0) | 37,419 | (38.2) | 9 | (0.0) | 1,069 | (1.1) |
| 35-39 | 72,618 | 50,544 | (69.6) | 16,540 | (22.8) | 33,060 | (45.5) | 27 | (0.0) | 917 | (1.3) |
| 40-44 | 50,902 | 39,072 | (76.8) | 11,455 | (22.5) | 26,627 | (52.3) | 34 | (0.1) | 956 | (1.9) |
| 45-49 | 38,885 | 32,563 | (83.7) | 8,234 | (21.2) | 22,950 | (59.0) | 97 | (0.2) | 1,282 | (3.3) |
| 50-54 | 36,432 | 32,330 | (88.7) | 6,491 | (17.8) | 23,677 | (65.0) | 263 | (0.7) | 1,899 | (5.2) |
| 55-59 | 35,192 | 32,304 | (91.8) | 5,048 | (14.3) | 23,993 | (68.2) | 616 | (1.8) | 2,647 | (7.5) |
| 60-64 | 26,358 | 24,854 | (94.3) | 2,860 | (10.9) | 17,665 | (67.0) | 1,212 | (4.6) | 3,117 | (11.8) |
| 65-69 | 16,311 | 15,646 | (95.9) | 1,190 | (7.3) | 9,894 | (60.7) | 1,981 | (12.1) | 2,581 | (15.8) |
| 70-74 | 9,175 | 8,930 | (97.3) | 392 | (4.3) | 4,088 | (44.6) | 2,506 | (27.3) | 1,944 | (21.2) |
| 75-79 | 4,229 | 4,185 | (99.0) | 90 | (2.1) | 916 | (21.7) | 2,091 | (49.4) | 1,088 | (25.7) |
| ≥ 80 | 1,757 | 1,750 | (99.6) | 3 | (0.2) | 157 | (8.9) | 1,084 | (61.7) | 506 | (28.8) |

same starting age, those with CKM stages 1, 2, 3, and 4 had 1.5 (1.36 to 1.62), 3.7 (3.55 to 3.93), 4.5 (4.24 to 4.77), and 8.0 (7.37 to 8.68) fewer years of life expectancy, respectively, compared to CKM stage 0. When considering the number of CKM components, male participants with 1, 2, 3, 4, and 5 components had 1.5 (1.36 to 1.62), 3.7 (3.55 to 3.93), 4.5 (4.24 to 4.77), 8.0 (7.37 to 8.68), and 13.8 (12.07 to 15.45) fewer years of life expectancy, respectively, compared with those with no CKM components. Among female participants, the corresponding reductions in life expectancy were 2.9

**Table 3. Hazard ratios for risks of cause-specific mortality and end-stage kidney disease stratified by cardiovascular–kidney–metabolic syndrome stage.**

| CKM | All-cause mortality | | | | CVD mortality | | | Heart disease mortality | | | Stroke mortality | | |
|---|---|---|---|---|---|---|---|---|---|---|---|---|---|
| | N | N of deaths | HR* | (95% CI) | N of deaths | HR* | (95% CI) | N of deaths | HR* | (95% CI) | N of deaths | HR* | (95% CI) |
| Stage 0 | 147,024 | 3,002 | Ref. | | 282 | Ref. | | 65 | Ref. | | 98 | Ref. | |
| Stage 1 | 100,585 | 3,280 | 0.96 | (0.91, 1.02) | 368 | 1.13 | (0.95, 1.35) | 82 | 1.13 | (0.78, 1.63) | 116 | 0.95 | (0.70, 1.29) |
| Stage 2 | 238,647 | 22,469 | **1.36** | **(1.30, 1.42)** | 4,631 | **2.89** | **(2.51, 3.32)** | 1,211 | **3.35** | **(2.51, 4.46)** | 1,580 | **2.61** | **(2.07, 3.28)** |
| Stage 3 | 9,925 | 6,415 | **2.13** | **(2.02, 2.25)** | 1,594 | **5.27** | **(4.51, 6.16)** | 453 | **6.30** | **(4.59, 8.65)** | 538 | **4.54** | **(3.49, 5.89)** |
| Stage 4 | 19,421 | 6,423 | **2.37** | **(2.25, 2.49)** | 1,950 | **7.42** | **(6.40, 8.60)** | 557 | **9.50** | **(7.01, 12.87)** | 572 | **5.56** | **(4.33, 7.14)** |
| All CKM† | 368,578 | 38,587 | **1.33** | **(1.28, 1.39)** | 8,543 | **2.81** | **(2.45, 3.22)** | 2,303 | **3.29** | **(2.48, 4.37)** | 2,806 | **2.46** | **(1.96, 3.09)** |

| | DM mortality | | | | Kidney diseases mortality | | | Expanded CVD mortality†† | | | ESKD | | |
|---|---|---|---|---|---|---|---|---|---|---|---|---|---|
| | N | N of deaths | HR* | (95% CI) | N of deaths | HR* | (95% CI) | N of deaths | HR* | (95% CI) | N of ESKD | HR* | (95% CI) |
| Stage 0 | 147,024 | 11 | Ref. | | 15 | Ref. | | 308 | Ref. | | 53 | Ref. | |
| Stage 1 | 100,585 | 46 | **3.60** | **(1.66, 7.84)** | 21 | 0.92 | (0.42, 2.01) | 435 | **1.21** | **(1.03, 1.43)** | 63 | 1.38 | (0.9, 2.07) |
| Stage 2 | 238,647 | 1,196 | **22.00** | **(10.94, 44.25)** | 483 | **4.76** | **(2.66, 8.52)** | 6,310 | **3.62** | **(3.17, 4.13)** | 1,785 | **12.34** | **(9.14, 16.67)** |
| Stage 3 | 9,925 | 566 | **82.73** | **(40.63, 168.48)** | 246 | **15.09** | **(8.21, 27.74)** | 2,406 | **8.00** | **(6.92, 9.24)** | 477 | **56.91** | **(41.00, 79.00)** |
| Stage 4 | 19,421 | 515 | **68.79** | **(33.97, 139.30)** | 385 | **24.41** | **(13.52, 44.07)** | 2,850 | **10.36** | **(9.02, 11.90)** | 956 | **65.89** | **(48.34, 89.81)** |
| All CKM† | 368,578 | 2,323 | **20.19** | **(10.07, 40.48)** | 1,135 | **5.37** | **(3.03, 9.53)** | 12,001 | **3.53** | **(3.10, 4.02)** | 3,281 | **10.15** | **(7.54, 13.67)** |

*Hazard ratios were adjusted for age, sex, educational level, smoking status, drinking status, and physical activity.

†All CKM does not include stage 0.

††Expanded CVD criteria, which consisted of CVD plus diabetes and kidney diseases as a composite outcome.

CKM: cardiovascular–kidney–metabolic syndrome; N: number of participants; HR: hazard ratio; CI: confidence interval; CVD: cardiovascular disease; Ref: reference; DM: diabetes mellitus; ESKD: end-stage kidney disease.

(2.73 to 3.01), 5.5 (5.19 to 5.75), 6.9 (6.38 to 7.45), 9.9 (8.89 to 10.87), and 16.3 (14.61 to 18.00) years. On average, each additional CKM component was associated with a reduction in life expectancy of 2.5 years in males and 3.0 years in females (Fig 3).

We assessed the distribution of the number of CKM components (hypertension, CKD, diabetes, metabolic syndrome, and hyperlipidemia) present in our cohort (N = 515,602) (S6 Table). One-fifth (22.7%) of the participants had two or more CKM components. However, among those with any CKM component (N = 264,038), two-fifths (44.4%) had two or more components. Furthermore, one-ninth (11.7%) of participants in the entire cohort had three or more CKM components. However, among those with any CKM component, the proportion with three or more components increased to one-fifths (22.9%). Among participants aged ≥65 years with any CKM components, the multimorbidity became more common, with 69.9% had two or more components, 43.0% had three or more components, and 21.5% had four or more components.

We assessed participants' awareness of their chronic conditions based on whether they had sought medical treatment or received medication. Awareness rates were 54.6% for diabetes, 22.1% for hypertension – 130/80 mmHg, 43.4% for hypertension – 140/90 mmHg, and 3.3% for CKD. For hyperlipidemia, 2.9% were receiving medication for high triglycerides, and for metabolic syndrome, 38.3% were on medication for diabetes, hypertension, or high triglycerides (S7 Table).

As part of a comparison across different populations, we compared the prevalence rates of CKM stages between the U.S. National Health and Nutrition Examination Survey (NHANES) [31] and our cohort across different age groups (S8 Table). CKM stage 2 accounted for the majority of CKM cases, representing approximately 55% in U.S. NHANES and around 65% in our study cohort.

**Table 4. Hazard ratios for risks of cause-specific mortality and end-stage kidney disease stratified by number of cardiovascular–kidney–metabolic syndrome components.**

| | All-cause mortality | | | | CVD mortality | | | Heart disease mortality | | | Stroke mortality | | |
|---|---|---|---|---|---|---|---|---|---|---|---|---|---|
| | N | N of deaths | HR* | (95% CI) | N of deaths | HR* | (95% CI) | N of deaths | HR* | (95% CI) | N of deaths | HR* | (95% CI) |
| 0 Components | 251,564 | 6,750 | Ref. | | 761 | Ref. | | 174 | Ref. | | 242 | Ref. | |
| 1 Component | 146,826 | 11,065 | 1.21 | (1.17, 1.26) | 2,190 | 2.02 | (1.84, 2.22) | 537 | 2.16 | (1.78, 2.62) | 746 | 2.15 | (1.83, 2.53) |
| 2 Components | 56,647 | 8,772 | 1.49 | (1.43, 1.54) | 2,084 | 2.84 | (2.58, 3.13) | 547 | 3.38 | (2.78, 4.11) | 711 | 2.94 | (2.48, 3.48) |
| 3 Components | 40,047 | 7,279 | 1.57 | (1.51, 1.63) | 1,816 | 3.17 | (2.88, 3.50) | 490 | 3.72 | (3.05, 4.53) | 603 | 3.18 | (2.68, 3.77) |
| 4 Components | 16,001 | 5,406 | 2.12 | (2.03, 2.21) | 1,429 | 4.54 | (4.10, 5.02) | 433 | 6.24 | (5.09, 7.64) | 432 | 4.12 | (3.43, 4.94) |
| 5 Components | 4,517 | 2,317 | 3.53 | (3.34, 3.72) | 545 | 6.68 | (5.88, 7.58) | 187 | 10.53 | (8.32, 13.34) | 170 | 6.23 | (4.97, 7.80) |
| Increase by 1 component | | | 1.22 | (1.21, 1.23) | | 1.37 | (1.35, 1.40) | | 1.48 | (1.44, 1.53) | | 1.33 | (1.29, 1.37) |

| | DM mortality | | | | Kidney diseases mortality | | | Expanded CVD mortality† | | | ESKD | | |
|---|---|---|---|---|---|---|---|---|---|---|---|---|---|
| | N | N of deaths | HR* | (95% CI) | N of deaths | HR* | (95% CI) | N of deaths | HR* | (95% CI) | N of ESKD | HR* | (95% CI) |
| 0 Components | 251,564 | 62 | Ref. | | 41 | Ref. | | 864 | Ref. | | 123 | Ref. | |
| 1 Component | 146,826 | 238 | 3.18 | (2.27, 4.44) | 154 | 2.58 | (1.73, 3.86) | 2,582 | 2.14 | (1.96, 2.34) | 374 | 3.83 | (3.05, 4.81) |
| 2 Components | 56,647 | 370 | 8.50 | (6.15, 11.76) | 258 | 6.52 | (4.42, 9.62) | 2,712 | 3.38 | (3.10, 3.70) | 660 | 14.80 | (11.89, 18.42) |
| 3 Components | 40,047 | 507 | 14.40 | (10.47, 19.80) | 235 | 7.45 | (5.04, 11.00) | 2,558 | 4.05 | (3.71, 4.43) | 632 | 19.36 | (15.55, 24.11) |
| 4 Components | 16,001 | 620 | 33.37 | (24.27, 45.88) | 283 | 15.72 | (10.65, 23.19) | 2,332 | 6.72 | (6.14, 7.36) | 847 | 60.20 | (48.34, 74.96) |
| 5 Components | 4,517 | 537 | 116.68 | (84.64, 160.86) | 179 | 38.20 | (25.49, 57.24) | 1,261 | 14.16 | (12.80, 15.67) | 698 | 209.17 | (167.29, 261.54) |
| Increase by 1 component | | | 2.40 | (2.32, 2.49) | | 1.88 | (1.79, 1.97) | | 1.56 | (1.54, 1.58) | | 2.63 | (2.55, 2.71) |

Five components were hypertension, chronic kidney disease, diabetes, metabolic syndrome, and hyperlipidemia.

*HRs were adjusted for age, sex, educational level, smoking status, drinking status, and physical activity.

†Expanded CVD criteria, which consisted of CVD plus diabetes and kidney diseases as a composite outcome.

CKM: cardiovascular–kidney–metabolic syndrome; N: number of participants; HR: hazard ratio; CI: confidence interval; CVD: cardiovascular disease; Ref: reference; DM: diabetes mellitus; ESKD: end-stage kidney disease.

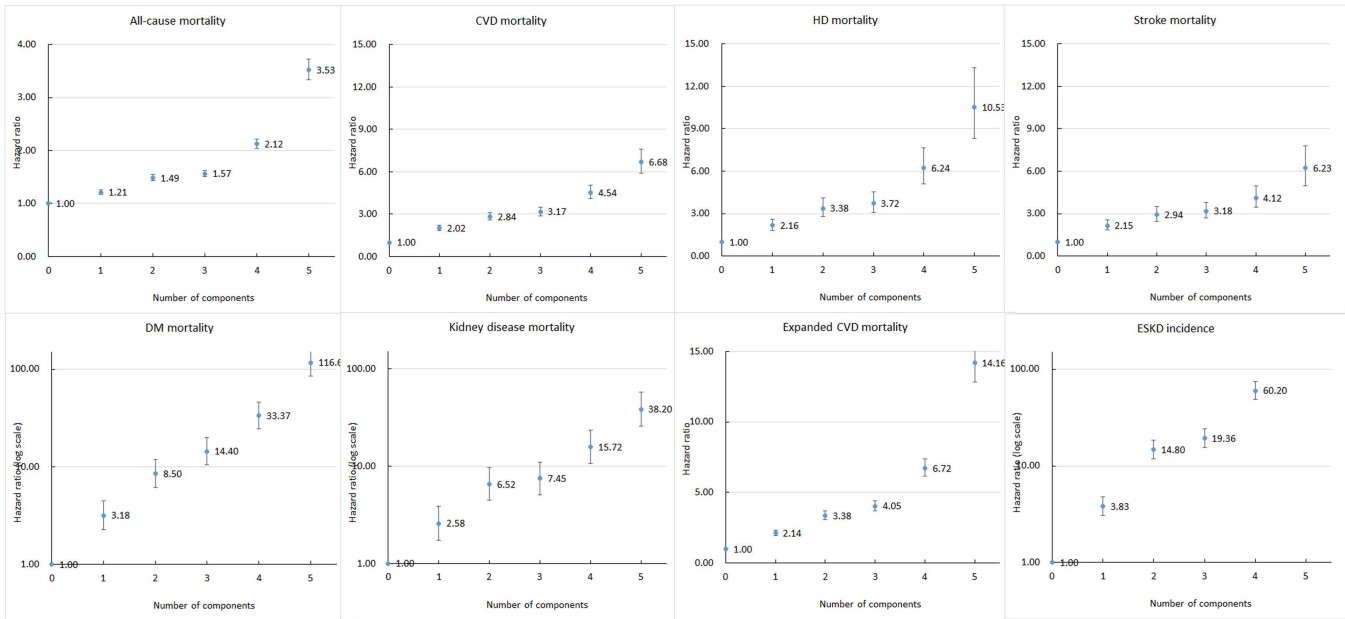

**Fig 2. Risks of cause-specific mortality and end-stage kidney disease stratified by number of components of cardiovascular–kidney–metabolic syndrome.** CVD: cardiovascular disease; HD: heart disease; DM: diabetes mellitus; ESKD: end-stage kidney disease. The points and whiskers represent the HRs and their 95% confidence intervals. The axes for diabetes mortality, kidney disease mortality, and ESKD incidence are presented on a logarithmic scale; the axes start at 1 instead of 0.

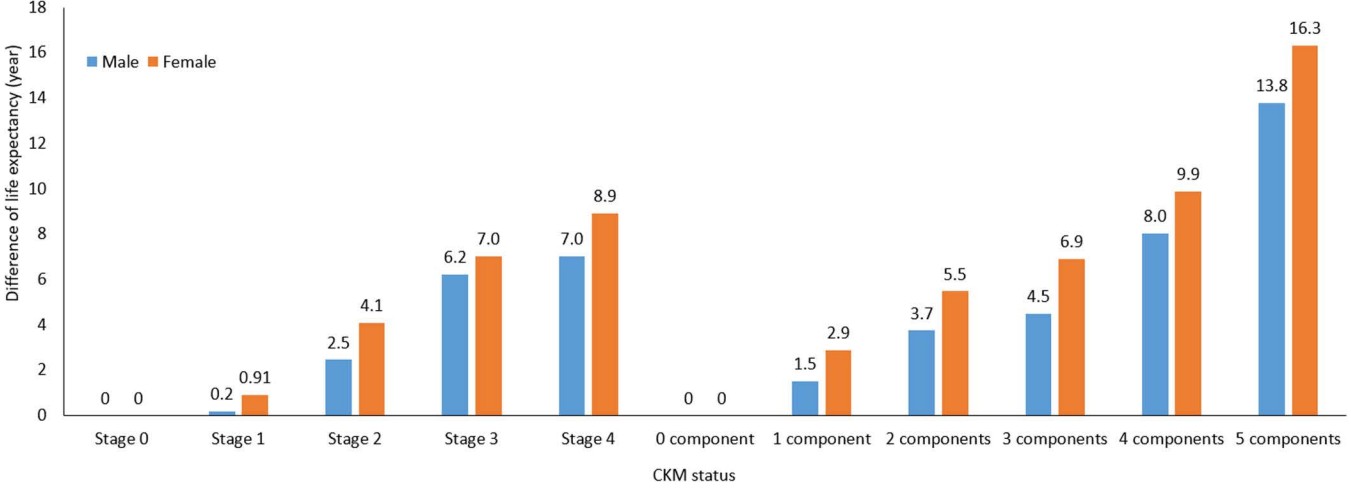

**Fig 3. Reduction in life expectancy by CKM status for male and female.** Stage 0 and 0 component served as the reference. CKM components included the presence of hypertension, chronic kidney disease, diabetes, metabolic syndrome, or hyperlipidemia. Differences in average remaining life expectancy as calculated using the life expectancy at the age of 30 years as the reference.

After accounting for both CKM prevalence and associated mortality risks, we estimated the PAF of CKM syndrome to be 18.7% (95% CI: 15.8–21.7) for all-cause mortality and 55.0% (95% CI: 49.0–60.4) for cardiovascular mortality (S9 Table).

To validate subgroup-specific patterns, the stratified analysis by sex and age groups still revealed a significant increase in the risk of all-cause mortality among those with CKM or more than two of its components (S10 Table).

To evaluate consistency and potential changes over time, we conducted an analysis among participants who had undergone two health examinations. The average follow-up duration between these two examinations was 2.5 years (mean (SD): 2.5 (2.4)). Approximately two-thirds (N = 148,639, 67.4%) of participants remained in the same CKM stage, while 18.3% (N = 40,320) progressed to a more severe stage, and 14.3% (N = 31,423) improved to a less severe stage (S11 Table). Spearman correlation analysis of CKM stages between the two time points showed a correlation coefficient of 0.62 (P < 0.0001). We incorporated a time-dependent analysis, which yielded similar results (S12 Table).

## Discussion

Seventy percent of the participants in our large Asian cohort had CKM. Among those aged ≥55 years, almost 90% had CKM. As the severity of CKM increased (starting from CKM Stage 2), the risks of all-cause mortality, CVD, and ESKD incidence also increased. When analyzing the effects of different combinations of the five chronic diseases, we determined that diabetes resulted in the highest risk of all-cause mortality, hypertension resulted in the highest risk of CVD, and CKD resulted in the highest risk of ESKD. However, when considering these five diseases as distinct components, we observed that the risks of all-cause mortality, CVD mortality, and ESKD increased with the number of CKM components present. Specifically, each additional component was associated with a 22% increase in the risk of all-cause mortality and a 37% increase in the risk of CVD mortality. Furthermore, each additional component reduced life expectancy by an average of 3 years.

A recent report from the U.S. NHANES [31] indicated a CKM prevalence of up to 90%. We compared prevalence rates between these two studies across different age groups [31]. CKM stage 2 accounts for more than half of all CKM cases, with approximately 55% in the U.S. NHANES and about 65% in our study cohort. The proportion of CKM stage 1 is relatively lower in our study. The prevalence of obesity reported by the U.S. NHANES was 42.3% [32], whereas that in our cohort was 27.9% (using BMI 25 as the cutoff). This difference in obesity prevalence may have contributed to variations in CKM prevalence rates between the two studies. Both the U.S. NHANES and the present study indicate that more than half of the adult population has CKM stages 1–3, highlighting the substantial disease burden posed by CKM. The aforementioned findings indicate that the five components of CKM are interrelated. In this study, among participants with any of the five components, 44.4% had two or more components, 22.9% had three or more components, and 7.8% had four or more components. Similar results were reported in the U.S. [33], where 26.3% of individuals had CKD, diabetes, or CVD. Among these individuals, more than two of these conditions co-occurred in 30.6% of cases, indicating a clustering tendency among these diseases. These findings align with the new concept of CKM syndrome proposed by the AHA, which suggests a shift from focusing on individual risk factors to taking a holistic view of CKM [34]. Our results revealed that the risk of mortality increased with the number of CKM components observed. Although different risks contribute to various causes of death at varying levels, increased number of components was consistently associated with a higher risk of mortality. According to Medicare data in the United States, patients with both diabetes and CKD had higher risks of myocardial infarction, heart failure, renal-replacement therapy, and mortality than did those without diabetes or CKD [35]. The risk of mortality increases as one accumulates chronic conditions. For instance, individuals with both diabetes and CKD have a higher risk of mortality than do those with diabetes alone [36,37]. Similarly, the risk of mortality is higher in individuals with both diabetes and hypertension [38]. Prehypertension, when combined with diabetes, increases the risks of all-cause mortality and CVD mortality [39]. In addition, hypertension and CKD exert a synergistic effect. Regardless of whether the condition is CKD or diabetic kidney disease, the presence of hypertension is associated with an increased risk of CVD [40–42].

An important question regarding CKM syndrome is whether it is better than current practice for identifying high-risk individuals. The most significant difference between CKM syndrome and previous frameworks that emphasized CVD risk factors is the inclusion of CKD. We estimated the PAF of CKM syndrome to be 18.7% (15.8–21.7) for all-cause mortality and 55.0% (49.0–60.4) for CVD mortality. CKD has historically been overlooked in CVD-attributable mortality [43], and

many reports have even suggested that CKD is overdiagnosed [44]. In our analysis, CKD also independently contributed a significant risk to CVD mortality even when controlling for blood pressure (130/80 mmHg), blood glucose (<126 mg/dL), absence of metabolic syndrome, and triglycerides (<135 mg/dL). These independent cases of CKD, accounting for approximately 28% of CKD participants, represent a high-risk population that should not be ignored. Since the PAF of CKD for CVD mortality is 7.6%, failing to include CKD in the definition of CKM syndrome could lead to the missed attribution of approximately 11% of CVD deaths (7.6/(76.2–7.6) = 11%). When considering the actual number of deaths, the number of individuals with CKD who die from CVD is nearly half that of all individuals with hypertension (130/80 mmHg). Therefore, incorporating CKD into the new CKM definition would account for a greater number of CVD-related deaths. This figure is also nearly double the CVD mortality burden attributed to diabetes, highlighting that the newly defined CKM syndrome can effectively capture CKD-related mortality. In addition to causes of CVD mortality, CKD is also the leading risk factor for the development of ESKD. Incorporating CKD into the CKM definition not only captures its impact on cardiovascular outcomes but also highlights the importance of preventing progression to dialysis. This is particularly relevant for certain Asian countries, including Taiwan, where the prevalence and incidence of dialysis are among the highest in the world [45]. Emphasizing CKD prevention is of critical importance. The effect of CKD is substantial, yet awareness of the disease's effects at the population level is relatively low. Although evidence indicates an association of kidney disease with increased mortality and CVD risk, awareness of CKD remains inadequate, and current prevention strategies often fail to adequately address this low awareness [14,46]. Thus, redefining CKM can enhance our understanding of the effects of CKD on CVD and mortality risks, emphasizing the need for early CKD screening to mitigate these associated risks [46].

CKM consists of five chronic diseases, and the prevalence of these diseases, especially hypertension and CKD, increases with age. Moreover, the clustering of these components is more pronounced in the older population. Although the prevalence of these diseases was common in the older population, stratified analysis by age group still revealed a significant increase in the risk of all-cause mortality among older individuals with CKM or more than two of its components. Thus, the risk of mortality increases with the number of CKM components present, a finding that is particularly noteworthy given the rapid aging of populations worldwide. This result highlights the importance of providing integrated care for patients with these chronic diseases [31,47].

This study has several strengths. First, the study examines the relationship between CKM and various causes of mortality based on the AHA's updated definition. Using a large cohort and a follow-up period of more than 20 years, we explored various combinations of CKM stages and components and assessed their associated risks with different causes of mortality. Our analysis extended beyond CVD mortality and provided insights into mortality associated with diabetes and kidney diseases. Second, our cohort consisted of individuals who underwent general health examinations instead of patients who received diagnoses in hospitals, providing a snapshot that more accurately represents the distribution of CKM within a general population. Third, the study obtained comprehensive data on blood pressure, glucose, lipid profiles, renal function, and obesity-related indicators, which facilitated classification in accordance with the new CKM staging system. This study has some limitations that should be considered. First, we lacked data on physician-diagnosed diseases and medication use records, which could have led to incomplete information, particularly on factors such as peripheral artery disease and atrial fibrillation history in CKM stage 4. In addition, we acknowledged that anyone who takes a blood pressure-lowering drug may not have hypertension. We defined hypertensive participants as screened hypertension or under antihypertensive therapy. Second, we recruited participants who were undergoing health check-ups. Such individuals likely represent a healthier and socioeconomically advantaged segment of the population, thereby limiting the generalizability of our findings to the broader population. In addition, our cohort is relatively younger, which may affect the generalizability of the prevalence estimates. Third, our analysis was based only on baseline measurements, which do not account for changes in individual circumstances over time. Future research should focus on the impact of CKM stage changes over time to provide a more comprehensive understanding of its implications. However, we conducted an analysis among participants who had undergone two health examinations. Among them, we assessed the association between

changes in CKM stages and mortality risk. Approximately two-thirds of participants remained in the same CKM stage, while 18.3% progressed to a more severe stage, and 14.3% improved to a less severe stage. We incorporated a time-dependent analysis, which yielded similar results. However, given the relatively small subset of patients with repeated measurements, this additional longitudinal data point may lack sufficient power to detect true differences. Finally, this study focused exclusively on an Asian population, which may restrict the generalizability of our findings to populations of other ethnicities.

On the basis of the AHA's definition of CKM syndrome, this study examined the epidemiological distribution of CKM syndrome and its associations with various mortality outcomes. We examined the integration of multiple chronic disease components and found a cumulative effect of these diseases. Individuals with more components of CKM syndrome had higher risks of adverse outcomes, including all-cause mortality, CVD mortality, and ESKD. The PAFs of CKM syndrome were 18.7% for all-cause mortality and 55.0% for CVD mortality. We estimated that failing to include CKD in CKM syndrome could result in the missed attribution of approximately 11% of CVD deaths. These results indicate the importance of interdisciplinary research in understanding and addressing the complexities of disease syndromes such as CKM. Future studies should investigate the interrelated and multifaceted nature of CKM syndrome.

### Ethics statements

This study was approved by the Institutional Review Board of Taipei Medical University (Approval Number: N202404037). All participants gave written consent to analysis their unidentifiable data before being included in this study.

### Supporting information

**S1 Table. Restricted mean survival time regression model for all-cause and CVD mortality by cardiovascular–kidney–metabolic syndrome stage.**
(DOCX)

**S2 Table. Hazard ratios for all-cause and CVD mortality stratified by cardiovascular–kidney–metabolic syndrome stage (Multiple imputation imputed for missing information).**
(DOCX)

**S3 Table. Hazard ratios for all-cause and CVD mortality stratified by cardiovascular–kidney–metabolic syndrome stage (additionally considering the use of analgesics and cancer history).**
(DOCX)

**S4 Table.** (1) Hazard ratios for risks of cause-specific mortality and end-stage kidney disease stratified by number and type of cardiovascular–kidney–metabolic syndrome component. (2) Hazard ratios for risks of cause-specific mortality and end-stage kidney disease stratified by number and type of cardiovascular–kidney–metabolic syndrome component.
(DOCX)

**S5 Table. Remaining years of life stratified by CKM stage and number of components for male and female.**
(DOCX)

**S6 Table. Prevalence of CKM components in study cohort stratified by age.**
(DOCX)

**S7 Table. Prevalence of self-reported medication and hazard ratios with risk of all-cause mortality by CKM components.**
(DOCX)

**S8 Table. Prevalence of cardiovascular–kidney–metabolic syndrome in this cohort and in the US National Health and Nutrition Examination Survey.**
(DOCX)

**S9 Table. Population-attributable fractions of cardiovascular–kidney–metabolic syndrome stages and components for all-cause and cardiovascular-disease mortality.**
(DOCX)

**S10 Table. Age group-specific and sex-specific risks of all-cause mortality stratified by cardiovascular–kidney–metabolic syndrome status.**
(DOCX)

**S11 Table. The distribution of baseline cardiovascular–kidney-metabolic syndrome status and follow-up CKM status among participants with a second visit.**
(DOCX)

**S12 Table. Hazard ratios with risk of all-cause mortality stratified by cardiovascular–kidney–metabolic syndrome status among participants with 2nd visit, Time-dependent results.**
(DOCX)

**S1 Fig. Participants selection process.**
(DOCX)

**S2 Fig. Kaplan–Meier survival curves stratified by cardiovascular–kidney–metabolic syndrome stage.**
(DOCX)

**S3 Fig. Kaplan–Meier survival curves stratified by cardiovascular–kidney–metabolic syndrome components.**
(DOCX)

**S4 Fig. Age and education-adjusted CKM prevalence.**
(DOCX)

**S5 Fig. Risks of cause-specific mortality and end-stage kidney disease stratified by cardiovascular–kidney–metabolic syndrome stage.**
(DOCX)

**S6 Fig. Age-specific prevalence of hypertension, chronic kidney disease, diabetes, metabolic syndrome, and hypertriglycerides in study cohort.**
(DOCX)

**S1 File. STROBE statement.**
(DOCX)

**S2 File. SAS code for life-table methods.**
(DOCX)

## Acknowledgments

We are grateful to the Health and Welfare Data Science Center and National Health Research Institutes for providing administrative and technical support.

## Author contributions

**Conceptualization:** Min-Kuang Tsai, Juliana Tze-Wah Kao, Chung-Shun Wong, Chia-Te Liao, Wei-Cheng Lo, Kuo-Liong Chien, Chi-Pang Wen, Mai-Szu Wu, Mei-Yi Wu.

**Data curation:** Min-Kuang Tsai, Mei-Yi Wu.

**Formal analysis:** Min-Kuang Tsai.

**Investigation:** Min-Kuang Tsai, Juliana Tze-Wah Kao, Chung-Shun Wong, Chia-Te Liao, Wei-Cheng Lo, Kuo-Liong Chien, Chi-Pang Wen, Mai-Szu Wu, Mei-Yi Wu.

**Methodology:** Min-Kuang Tsai, Juliana Tze-Wah Kao, Chung-Shun Wong, Chia-Te Liao, Wei-Cheng Lo, Kuo-Liong Chien, Chi-Pang Wen, Mai-Szu Wu, Mei-Yi Wu.

**Project administration:** Min-Kuang Tsai, Chia-Te Liao, Mei-Yi Wu.

**Resources:** Chia-Te Liao.

**Software:** Min-Kuang Tsai.

**Supervision:** Min-Kuang Tsai, Juliana Tze-Wah Kao, Chung-Shun Wong, Chia-Te Liao, Kuo-Liong Chien, Chi-Pang Wen, Mai-Szu Wu, Mei-Yi Wu.

**Validation:** Min-Kuang Tsai, Juliana Tze-Wah Kao, Chung-Shun Wong, Chia-Te Liao, Wei-Cheng Lo, Kuo-Liong Chien, Chi-Pang Wen, Mai-Szu Wu, Mei-Yi Wu.

**Visualization:** Min-Kuang Tsai.

**Writing – original draft:** Min-Kuang Tsai, Juliana Tze-Wah Kao, Chung-Shun Wong, Chia-Te Liao, Kuo-Liong Chien, Chi-Pang Wen, Mai-Szu Wu, Mei-Yi Wu.

**Writing – review & editing:** Min-Kuang Tsai, Juliana Tze-Wah Kao, Chung-Shun Wong, Chia-Te Liao, Wei-Cheng Lo, Kuo-Liong Chien, Chi-Pang Wen, Mai-Szu Wu, Mei-Yi Wu.

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
