## [Editor Report · Decision Letter 0]

13 Dec 2024

Dear Dr Wu,

Thank you for submitting your manuscript entitled "Unveiling the Creation of Cardiovascular–Kidney–Metabolic Syndrome: the Contribution to All-Cause and Cardiovascular Mortality - A cohort of more than half-a-million participants" for consideration by PLOS Medicine.

Your manuscript has now been evaluated by the PLOS Medicine editorial staff as well as by an academic editor with relevant expertise and I am writing to let you know that we would like to send your submission out for external peer review.

Please re-submit your manuscript within two working days, i.e. by Dec 17 2024.

Feel free to email me at atosun@plos.org or us at plosmedicine@plos.org if you have any queries relating to your submission.

Kind regards,

Alexandra Tosun, PhD

Associate Editor

PLOS Medicine

---

## [Decision Letter · Decision Letter 1]

23 Jan 2025

Dear Dr Wu,

Many thanks for submitting your manuscript "Unveiling the Creation of Cardiovascular–Kidney–Metabolic Syndrome: the Contribution to All-Cause and Cardiovascular Mortality - A cohort of more than half-a-million participants" (PMEDICINE-D-24-04247R1) to PLOS Medicine. The paper has been reviewed by subject experts and a statistician; their comments are included below and can also be accessed here: [LINK]

As you will see, the reviewers express an interest in the study, but there are a number of statistical concerns and points that need to be clarified. After discussing the paper with the editorial team and an academic editor with relevant expertise, I'm pleased to invite you to revise the paper in response to the reviewers' comments. We plan to send the revised paper to some or all of the original reviewers, and we cannot provide any guarantees at this stage regarding publication.

We ask that you submit your revision by Feb 13 2025. However, if this deadline is not feasible, please contact me by email, and we can discuss a suitable alternative.

Don't hesitate to contact me directly with any questions (atosun@plos.org).

Best regards,

Alexandra

Alexandra Tosun, PhD

Associate Editor

PLOS Medicine

atosun@plos.org

Comments from the academic editor:

My impression is that there are mixed feelings about the CKM syndrome and a sense of skepticism about the usefulness of AHA-defined syndromes in the clinical community. It will be important for the authors to make clear how the conceptual model of this type of research helps us to improve health care and health. The CKM stages are defined by progressively more severe organ dysfunction - was there any doubt about the findings? The authors should discuss the novelty of their findings more clearly, e.g. one could argue that it is well known that risks accumulate so that they increase as a person has more diseases. The aim of the paper should also be communicated more clearly, whether it is to predict mortality or disease events, to create clusters of patient types, or to investigate causal inference. In short, the manuscript would benefit from a stronger presentation with clearer rationale and goals.

Comments from the editorial team:

We request that you include a more detailed description of the study cohort in the main text of the manuscript, as we feel that the references to the previous reports are not sufficient.

Comments from the reviewers:

Reviewer #1: Following the recent American Heart Association (AHA) publication defining Cardiovascular-Kidney-Metabolic (CKM) Syndrome, this research aimed to use the AHA CKM syndrome definition to understand both prevalence of CKM and associations between CKM severity and serious outcomes in an Asian cohort. The authors primarily used Cox regression adjusted for baseline characteristics to understand the associations between various operationalisations of CKM (e.g., CKM severity stages 1-4, types of diseases constituting CKM, number of CKM diseases present) and various mortality endpoints plus end-stage kidney disease. They found most of the population examined had some level of CKM and, perhaps unsurprisingly, increasing severity of CKM led to worse outcomes.

This is an interesting paper that is broadly well presented and important in the context of CKM syndrome emerging as a new biomedical and public health concept. I have concerns about the lack of methodological detail provided - particularly important when analysing observational cohorts - and the bulk of my feedback below seeks to elicit this detail, but I am confident this is fully addressable by the authors. I am more troubled by a substantial limitation that I feel has not yet been sufficiently described and addressed by the authors. The issue is all analyses appear to be based on baseline data. That is, if a participant was recruited when they were healthy and had no CKM-defining disease states, they were assumed to be disease-free until they had an endpoint recorded. Similarly for patients having more (or even less) severe CKM over time. There is an obvious risk of misclassification bias as patients are followed-up over the long term to death. I have provided more detail on this below, but this must be called out early and a cogent argument presented to convince readers the analyses presented are valid.

Detailed feedback on the manuscript is provided below. My primary focus has been methodological, but I have made a few broader points as well. All items are major unless indicated otherwise. While I avoid "nitpicking" language used, there were a couple of cases (1 & 2 below) where I was troubled by the words chosen in prominent positions, felt they may be misleading, and thought it worth noting.

1. [MINOR] The title wording is confusing to me. Suggest rewording - especially reconsider the use of "Unveiling the creation…" as this doesn't really make sense in the context of the paper.

2. [MINOR] Similarly, the use of the word "mitigate" in the first sentence of the abstract is confusing.

3. [MINOR] Please ensure all abbreviations are defined when first used. "CKM" is used in the second para of the Introduction but is not defined until the third para.

4. The Introduction would benefit from being more focussed on defining CKM syndrome and establishing the motivation for this research clearly and succinctly. The first paragraph is fine, but the second seems to be extraneous to the purpose of this paper i.e., quantifying the prevalence and mortality associations of CKM syndrome in a large observational cohort does not require the historical development of CKM. I would also reword the third paragraph to be clearer, more focussed, and as succinct as possible e.g., "In November 2023, the American Heart Association (AHA) published new guidelines that defined cardiovascular-kidney-metabolic (CKM) syndrome for the first time. CKM syndrome is defined and staged based on the presence of clinical characteristics (e.g. obesity) and comorbidities (e.g., CVD, CKD). Because CKM syndrome is a newly defined concept…" Note I am not suggesting the authors must use this example wording, rather I hope it provides a helpful illustration of one way to improve clarity.

5. I appreciate cohort details have been reported previously, but I think a little more detail is needed in this paper. Specifically, how were participants recruited, when were the tests and questionnaires completed (once at baseline? repeatedly throughout follow-up?), characterising of follow-up (at least mean follow-up), and was there any in-person follow-up or was follow-up only done by linkage to deaths?

6. Relatedly, how do the analyses account for changes in an individual's health status over time e.g., a patient who changes CKM stage due to development of a disease like hypertension, CKD, etc.?

7. Please define CKM Stage 0 because the authors use this later in the paper. Consider including in the paper/supplement something similar to "Table 3. Definitions of CKM Health Stages" from the paper's reference 20. Being able to see the CKM syndrome stages easily in a table is very helpful.

8. Relatedly, it's extremely important to be clear on whether the authors are using the AHA's definitions or if they are adapting the AHA's definitions for this research. For example, in reference 20 the AHA includes "or HbA1c between 5.7% and 6.4%" in their definition of Stage 1 but it is omitted from the Stage 1 definition in this paper. I am not sure if this is a typo or if the authors are adapting the AHA's definition for clinical or logistical reasons.

9. How did the authors account for the use of antihypertensives for indications outside of hypertension e.g., migraine prophylaxis?

10. Why was reference 26 included? It appears unrelated to the sentence it was cited in.

11. [MINOR] While hazard ratios are the generally accepted statistic for time to event analyses, they have some limitations (refer DOIs: 10.1093/ehjacc/zuae017 and 10.1056/EVIDe2300142). Consider supplementing the key HR outcomes with another statistic like difference in median survival or RMST.

12. When making statements about significance (e.g., "However, there was no significant difference in the risk of other causes of death between CKM Stage 1 and Stage 0 participants.") then appropriate statistics should be presented to justify this claim such as a p-value for the statistical contrast testing this specific assertion.

13. "We examined the risk of death by comparing the remaining years of life expectancy in male and female participants." How was life expectancy estimated? This appears to be related to "Chiang's life table method" but reference 28 did not have sufficient detail about how this method is defined and used, rather it appeared to be another application of the method. Please ensure the techniques used and assumptions made supporting these analyses are fully detailed in the Methods so they can be replicated. This is particularly important when using Cox regression because Cox PH makes no assumptions about baseline hazard.

14. I'd encourage the authors to provide their SAS code as a supplement, particularly for the more novel techniques like estimating and comparing life expectancy/years of life lost.

15. "Among participants with all five components, 44.4% had at least two components, 22.9% had at least three components, and 7.8% had at least four components." My apologies but I do not understand this sentence at all and am struggling with this Results paragraph more generally. Suggest rewording here and in the Discussion.

16. [MINOR] "We examined the risk of individual CKM components (Table 3)." I think the results in this para are from Table S5-1, not Table 3?

17. [MINOR] "Although the relative hazard ratios of different chronic disease risks for mortality varied across different causes of death, we found that with an increasing number of components, the risk of death also increased. This finding demonstrates a dose-response relationship across all-causes of death and the incidence of ESKD." Consider putting sentences like this that interpret the results in the Discussion rather than the Results.

18. "We assessed participants' awareness of their chronic conditions based on whether they had sought medical treatment or received medication." This needs to be covered fully in the Methods before results are presented, noting the results presented are quite alarming! I think it's also worth being clear in the Introduction why questions of participant awareness are important to be considered as part of this particular study.

19. The authors use of the word "clustering" (e.g., "Our results revealed that CKM component clustering is positively associated with the risks of CVD mortality, all-cause mortality, diabetes mortality, and ESKD…") is a bit confusing because clustering has specific meanings in statistics and data science. Consider replacing with more precise language like "the risk of mortality increased with the number of CKM components observed" rather than using "clustering" as shorthand.

20. PAF results are introduced for the first time in the Discussion. Please describe how PAF was calculated in the Methods and present results in the Results section before discussing in the Discussion section.

21. "Such individuals likely represent a healthier and socioeconomically advantaged segment of the population, thereby limiting the generalizability of our findings to the broader population. However, the use of relative risks (HRs) in our analysis helps mitigate this limitation by allowing for comparisons within groups." Hazard ratios (or any other statistic) do not assist with generalizability of results if the sample itself is biased as described. Also, hazard ratios are most definitely not relative risks. Please reword.

22. "Third, our analysis was based only on baseline measurements, which do not account for changes in individual circumstances over time." Ah! This answers my earlier questions. This is a very serious limitation. This should have been made clearer much earlier in the Methods and justification provided as to the validity of the findings in the face of this limitation.

23. "However, sensitivity analyses using data from participants who underwent two examinations yielded similar results (Table S8)." I do not think this analysis mitigates the limitation identified. All CKM components identified (hypertension, CKD, diabetes, metabolic syndrome, and hyperlipidaemia) emerge and change over time, and this ostensibly simple analysis does not mitigate this clinical reality. Also, again, these results presented in the Discussion were not previously included in the Methods or Results.

Reviewer #2: The paper entitled: "Unveiling the Creation of Cardiovascular-Kidney-Metabolic Syndrome: the Contribution to All-Cause and Cardiovascular Mortality - A cohort of more than half-a-million participants" was aimed to clarify the role of cardiovascular-kidney-metabolic (CKM) syndrome as risk factor for the of all-cause and cardiovascular mortality, and end-stage kidney disease (ESKD). For this purpose, the authors analyzed a cohort of 515,602 individuals that were followed for almost 25 years. The CKM syndrome was documented in 71.5% of the study population, the age increased the prevalence of the CKM syndrome. CKM syndrome increased the risk of all-cause and cardiovascular mortality, and ESKD. Basically, this is an interesting analysis, with an important epidemiological implication about cardiovascular risk. Discussion is focused on the topic of the paper with a correct description of the strengths and limitations of the study. However, the paper presents few weaknesses. Therefore, I invite the authors to address the criticisms that I raised:

Major comments

1) The classification of the severity of the CKM syndrome should be consistent with to that reported by the AHA, mentioned in the reference #19.

2) The Authors should report in all tables and figures how are presented the results (mean ± SD, percentages ….).

Minor comments:

1) The title should be simplified. I think that "The Cardiovascular-Kidney-Metabolic Syndrome: the Contribution to All-Cause and Cardiovascular Mortality" is enough.

2) The English should be accurately checked.

3) The introduction can be shortened. History of CKM syndrome can be erased.

Reviewer #3: 1. The authors only mentioned that about 0.5 million participants were included from a health screening program. However, it is not clear who the participants were or how they were selected. Was participation in health screening program voluntary or mandatory? Were there specific inclusion and exclusion criteria? How well do the participants represent the general population or the target population of interest? What are the scope and purpose of the health screening program, and have these factors influenced the study population toward a healthier or less healthy group? Without this information, it is unclear whether there is potential bias in the study.

2. The prevalence of CKM is unexpectedly high. Even in the youngest age group of 20-24 years, nearly 50% of individuals had prevalent CKM at baseline. Could the authors provide any explanations? Is this prevalence comparable to those reported in other studies?

3. The study population is very young, with a mean age of 40 years at baseline. How might this affect the generalisability of the study findings to the general population?

4. The definition of CKM uses several laboratory measurements, and individuals with missing data were excluded from the analysis. How do these missing data affect the prevalence of CKM and its association with mortality? Have the authors considered using other methods to handle the missing data?

5. A description of baseline characteristics of participants is needed in the result section.

6. Whether other severe medical conditions, such as cancer or other chronic diseases, were considered as confounders or excluded from the analysis?

---

* Please upload any figures associated with your paper as individual TIF or EPS files with 300dpi resolution at resubmission; please read our figure guidelines for more information on our requirements: http://journals.plos.org/plosmedicine/s/figures. While revising your submission, please upload your figure files to the PACE digital diagnostic tool, https://pacev2.apexcovantage.com/. PACE helps ensure that figures meet PLOS requirements. To use PACE, you must first register as a user. Then, login and navigate to the UPLOAD tab, where you will find detailed instructions on how to use the tool. If you encounter any issues or have any questions when using PACE, please email us at PLOSMedicine@plos.org.

* FINANCIAL DISCLOSURES: The funding statement should include: specific grant numbers, initials of authors who received each award, URLs to sponsors’ websites. Also, please state whether any sponsors or funders (other than the named authors) played any role in study design, data collection and analysis, the decision to publish, or preparation of the manuscript. If they had no role in the research, include this sentence: “The funders had no role in study design, data collection and analysis, decision to publish, or preparation of the manuscript.”

* COMPETING INTEREST: All authors must declare their relevant competing interests per the PLOS policy, which can be seen here: https://journals.plos.org/plosmedicine/s/competing-interests

For authors with ties to industry, please indicate whether any of the interests has a financial stake in the results of the current study.

* ETHICS STATEMENTS: Please provide the approval number and details on the consent procedure (e.g. written, oral or waived by the ethics committee).

FIGURES AND TABLES

SUPPLEMENTARY MATERIAL

REFERENCES

* Where website addresses are cited, please include the complete URL and specify the date of access (e.g. [accessed: 12/06/2024]).

STUDY TYPE-SPECIFIC REQUESTS

* Abstract: Please include the study design, population and setting, number of participants, years during which the study took place (enrollment and follow up), length of follow up, and main outcome measures.

* Please ensure that the study is reported according to the STROBE (or appropriate STOBE extension) guideline (available from: https://www.equator-network.org/reporting-guidelines/strobe) and include the completed STROBE (or STROBE extension) checklist as Supporting Information. Please add the following statement, or similar, to the Methods: "This study is reported as per the Strengthening the Reporting of Observational Studies in Epidemiology (STROBE) guideline (S1 Checklist)." When completing the checklist, please use section and paragraph numbers, rather than page numbers.

* For all observational studies, in the manuscript text, please indicate: (1) the specific hypotheses you intended to test, (2) the analytical methods by which you planned to test them, (3) the analyses you actually performed, and (4) when reported analyses differ from those that were planned, transparent explanations for differences that affect the reliability of the study's results. If a reported analysis was performed based on an interesting but unanticipated pattern in the data, please be clear that the analysis was data driven.

* Please state in the Methods section whether the study had a prospective protocol or analysis plan. If a prospective analysis plan (from your funding proposal, IRB or other ethics committee submission, study protocol, or other planning document written before analyzing the data) was used in designing the study, please include the relevant document(s) with your revised manuscript as a Supporting Information file to be published alongside your study and cite it in the Methods section. A legend for this file should be included at the end of your manuscript. If no such document exists, please make sure that the Methods section transparently describes when analyses were planned, and when/why any data-driven changes to analyses took place. Changes in the analysis, including those made in response to peer review comments, should be identified as such in the Methods section of the paper, with rationale.

---

## [Decision Letter · Decision Letter 2]

6 Mar 2025

Dear Dr Wu,

Many thanks for re-submitting your manuscript "The Cardiovascular-Kidney- Metabolic Syndrome: the Contribution to All-Cause and Cardiovascular Mortality" (PMEDICINE-D-24-04247R2) to PLOS Medicine. The paper has been seen again by a subject expert and a statistician; their comments are included below and can also be accessed here: [LINK]

As you will see, the statistical reviewer still has concerns about the manuscript, that require further clarification. Please carefully address the comments in a further revision. We plan to send the revised paper to some or all of the original reviewers.

We ask that you submit your revision by Mar 27 2025. However, if this deadline is not feasible, please contact me by email, and we can discuss a suitable alternative.

Don't hesitate to contact me directly with any questions (atosun@plos.org).

Best regards,

Alexandra

Alexandra Tosun, PhD

Associate Editor

PLOS Medicine

atosun@plos.org

Comments from the reviewers:

Reviewer #1: Thanks to the authors for the clear and cogent responses to my feedback and associated changes to the manuscript. In most cases I agree and consider the items closed (assume this is the case unless otherwise indicated). I have identified some follow-up points for further consideration that I have detailed below.

Point 6

Minor point first. I think supplementary table S12 has been mislabelled as S13, and I am assuming the content in the current "S13" pertains to this point.

Looking at the left half of the table, I don't quite understand the value of adding analyses based only on the second set of results of those patients who underwent two health examinations. There is no particular reason to believe these results would be different to the main results except for a different distribution of patients across the CKM levels, assuming patients increase CKM stages over time. I suggest this half of the table is removed or the further reasoning provided. More informative would be to report the change profile of patients with two measures. How separated in time were the first and second measures on average? Did patients tend to stay in the same CKM stage or change stages? If they changed stages, did they go up or down? By how many levels?

The inclusion of time-dependent analyses as the right half of the table makes more sense and I would suggest it remains in the supplement. However, having only one extra longitudinal data point for a relatively small subset of patients is unlikely to be powered to detect any true difference, and I suggest this is briefly acknowledged as a limitation.

I remain concerned that there is a separation of ~16.5 years between ascertainment of CKM stage and end of follow up. That said, I appreciate it is not possible to rectify this issue with the existing dataset. Given the recency of CKM emerging as a topic of research interest, I acknowledge the value of having these results in the literature as a basis for future research despite this limitation.

Point 8

Noting the authors have defined CKM levels for the purpose of this research in Table S1, I think it remains important to identify each difference between the AHA's definitions and the definitions in this paper and to clearly explain why in each case. This could be in the main paper or in the supplementary materials.

Point 9

I don't the authors have quite understood my point. Blood pressure lowering medicines can be used for many indications outside of hypertension e.g., fluid retention, migraine prophylaxis, anxiety. This means it is risky to run the logic of "anyone who takes a blood pressure lowering drug has hypertension" because you will flag people using blood pressure lowering drugs for non-hypertension indications as having hypertension. Perhaps the best approach here is one of clarification and acknowledging this as a limitation? So, for example, rather than saying "Hypertension" in Table 1, say "On antihypertensive therapy" or make it clear in the table notes.

Point 12

This is now OK and required no further action. That said, please take extreme care when relying on overlap or separation of confidence intervals for the purpose of "significance". Point 21 in this publication has a nice explanation https://link.springer.com/article/10.1007/s10654-016-0149-3

Point 18

I think the authors missed the last sentence in my comment: "I think it's also worth being clear in the Introduction why questions of participant awareness are important to be considered as part of this particular study."

Point 19

Very minor but suggest using the word "associated" rather than "correlated" in this context because correlation is a specific statistical concept.

Point 20

Thank you for providing a definition for PAF in your methods. Please also ensure any results you want to discuss in the Discussion are presented first in the Results. This applies to all results, not just PAF.

Reviewer #3: The authors have addressed my concerns properly.

---

## [Decision Letter · Decision Letter 3]

15 Apr 2025

Dear Dr. Wu,

Thank you very much for re-submitting your manuscript "The Cardiovascular-Kidney- Metabolic Syndrome: the Contribution to All-Cause and Cardiovascular Mortality" (PMEDICINE-D-24-04247R3) for review by PLOS Medicine.

Thank you for your detailed response to the reviewer's comments. I have discussed the paper with my colleagues, and it has also been seen again by the statistical reviewer. The changes made to the paper were satisfactory to the reviewer. There are a number of editorial comments that require attention during a further revision. Please carefully revise the manuscript according to the editors' comments below. When submitting your revised paper, please once again include a detailed point-by-point response to the editorial comments.

[LINK]

In revising the manuscript for further consideration here, please ensure you address the specific points made by each reviewer and the editors. In your rebuttal letter you should indicate your response to the reviewers' and editors' comments and the changes you have made in the manuscript. Please submit a clean version of the paper as the main article file. A version with changes marked must also be uploaded as a marked up manuscript file. Please also check the guidelines for revised papers at http://journals.plos.org/plosmedicine/s/revising-your-manuscript for any that apply to your paper.

In addition to these revisions, you will need to complete some formatting changes, which you will receive in a follow up email. A member of our team will be in touch with a set of requests shortly..

We ask that you submit your revision within 1 week (Apr 22 2025). However, if this deadline is not feasible, please contact me by email, and we can discuss a suitable alternative.

Please do not hesitate to contact me directly with any questions (atosun@plos.org). If you reply directly to this message, please be sure to 'Reply All' so your message comes directly to my inbox.

We look forward to receiving the revised manuscript.

Sincerely,

Alexandra Tosun, PhD

Associate Editor 

PLOS Medicine

plosmedicine.org

Comments from Reviewers:

Reviewer #1: The authors have addressed all my comments, and I am happy to recommend acceptance. Well done and congratulations.

[LINK]

Requests from Editors:

GENERAL

* Please confirm that your title complies with to PLOS Medicine's style. Your title must be nondeclarative and not a question. It should begin with main concept if possible. "Effect of" should be used only if causality can be inferred, i.e., for an RCT. Please place the study design ("A randomized controlled trial," "A retrospective study," "A modelling study," etc.) in the subtitle (ie, after a colon).

* The funding statement should include: specific grant numbers, initials of authors who received each award, URLs to sponsors’ websites. Also, please state whether any sponsors or funders (other than the named authors) played any role in study design, data collection and analysis, the decision to publish, or preparation of the manuscript. If they had no role in the research, include this sentence: “The funders had no role in study design, data collection and analysis, decision to publish, or preparation of the manuscript.”

* Data availability statement: If available, please also provide an email address that provides a clear point of contact for researchers interested in the data. Please update the metadata in the online submission form with the details provided on page 18.

* Please include the statement on code availability in the data availability statement.

* Please include the completed STROBE checklist as Supporting Information. Please add the following statement, or similar, to the Methods: "This study is reported as per the Strengthening the Reporting of Observational Studies in Epidemiology (STROBE) guideline (S1 Checklist)."

* Please ensure that all abbreviations are defined at first use throughout the text (including statistical abbreviations). Please also check figures and tables.

* Please review your text for claims of novelty or primacy (e.g. 'for the first time', ‘pioneering’) and remove this language.

* Please check that any use of statistical terms (such as trend or significant) are supported by the data, and if not please remove them.

* Please ensure that tables and figures, including those in supplementary files, are appropriately referenced in the main text.

* Statistical reporting: Please revise throughout the manuscript, including tables and figures.

a) Please report statistical information as follows to improve clarity for the reader "22% (95% CI [13%,28%]; p</=)".

b) Please separate upper and lower bounds with commas instead of hyphens as the latter can be confused with reporting of negative values.

c) Please define statistical definitions at first use and repeat the abbreviated definitions (HR, CI etc.) for each set of parentheses.

* Since you are referring to "sex" throughout the manuscript, we suggest using the terms "male" and "female" instead of "men" and "women”. Please revise throughout.

ABSTRACT

* Please confirm that your abstract complies with our requirements, including providing all the information relevant to this study type https://journals.plos.org/plosmedicine/s/submission-guidelines#loc-abstract

* Please ensure that all numbers presented in the abstract are present and identical to numbers presented in the main manuscript text.

* In the Abstract, please include the important dependent variables that are adjusted for in the analyses.

* Please include the study design, setting, length of follow up, and baseline cohort demographics (e.g. average age, sex, race/ethnicity).

* l.30: We don't think the word "longitudinal" is appropriate here, since you didn't collect data at multiple points over time.

* l.31: We suggest removing the term "dose-response" as it does not seem appropriate in this context.

AUTHOR SUMMARY

* l.52: It is impossible for the lay reader to understand what "two components" means because you have not provided an explanation of the syndrome and its components. Also, it is not possible to understand the magnitude of the number two if you do not explain that there are five components until after the results.

INTRODUCTION

* Please ensure that the introduction ends with a clear description of the study question or hypothesis. We currently feel that the beginning and end of the last paragraph are somewhat repetitive.

METHODS AND RESULTS

* Please provide the ethical approval number and details on the consent procedure in the main text as done on page 18.

* l.140ff: Please consider moving Table S1 to the main text. We believe that the tabular presentation of the CKM stages is easier for the reader to follow.

* l.149-152: Please revise for grammar and syntax.

* l.225: When reporting ages, please ensure to provide a unit, such as years.

* In general, we feel that the results section needs to be improved in terms of detail, clarity and consistency. Throughout, there is a lack of reference to relevant tables/figures and the main tables and figures are not discussed in sufficient detail. Often the focus is on the analyses presented in the supplementary files, while the main results are described in only one or two sentences. Please consider whether moving some of the figures presented in the Supplementary Information to the main text would aid the reader.

* l.248ff: “Participants with CKM syndrome were more likely to be male (58.77%), have an educational level of middle school or lower (27.5%), current smoker (25.2%), and consume alcohol regularly (10.6%).” – It seems that this statement is not entirely correct. According to Table 1 participants with CKM syndrome were more likely to be non-smokers and non-drinkers. Please revise.

* Table 1/Figure 1: According to Table 1, we would have expected the prevalence of CKM to decrease with age (60+ years: 15.0%), whereas according to Figure 1, the prevalence increases with age. We believe that the calculations for Figure 1 are currently not 100% clear. Please check and revise and see our comment below.

* Figure 1: Please convert the stacked bar chart to another data representation, such as a table, and be sure to provide not only the percentages, but also the numerators and denominators. Please also add a unit for age (years).

* l.276: We suggest referencing Table S2 here before describing the results.

* l.291ff: Please include a reference to the relevant table showing the results on life expectancy (we assume Table S7). We find the description of the life expectancy results to be oversimplified. The numbers you present on line 292/293 seem to present only the results for age 30 (is this correct?). Also, we don't think it's appropriate to present ranges that mix results for males and females when you haven't presented a combined analysis. Overall, we don't think these results are presented appropriately or in sufficient detail. Please revise.

* l.298ff: Please note that the results for mortality risk based on the number of CKM components are not very clear. For example: "Among participants aged ≥65 years with all five CKM components, 69.9% had at least two components, 43.0% had at least three components, and 21.5% had at least four components". Do you mean with data available on all five components or between baseline and endline assessment? If an individual has all five components, it does not make sense to say that the individual has at least two, three, or four components. Please revise the entire paragraph for clarity.

* l.310ff: Since Table 3 is included in the main text, we think it's these results that should be discussed first in the Results section, rather than the results from Tables S6-1 and S6-2. Please revise.

* l.312: It appears that the numerical results for CVD mortality in the main text do not match the results in Table S6-1 (HR 2.42, 95%CI [1.84, 3.18] versus HR: 2.40; 95% CI: 2.18, 2.65). It appears that you used the results for "hypertension only" in the main text. Please check and revise.

* l.316: Please ensure to call out the relevant Table(s).

* l.322: Again, since Figure 2 is one of the main figures in your manuscript, we feel that you should explain the results in more detail.

* l.324: Please see our earlier comments (l.291ff) and revise accordingly.

* Table S8: Please explain the two groups for hypertension and ensure to describe the relevant group in the main text with sufficient detail.

* Table 1: Is there a reason you chose different age groups than in Figure 1? Why is there a tilde instead of a dash for BMI?

* Table 2: Please add a footnote explaining that "All CKM" does not include stage 0. Why is kidney-related mortality the only group that includes "related"? Have you explained anywhere what 'heart disease' includes?

* Table 3: Please see the comments for Table 2.

* Figure 2: Please indicate in the figure caption the meaning of the points and whiskers. Please show the axis beginning at zero. If this is not possible, please show a break in the axis.

* Figure 3: Please add that Stage 0/0 components served as the reference and add a brief explanation about the different components.

* Supplementary figures and tables: Please revise the supplementary figures and tables according to the above comments and make sure that the figure description is sufficiently detailed. In general, figures and tables should be self-explanatory on a stand-alone basis. Please revise accordingly.

DISCUSSION

* General guidance on organizing the Discussion: a short, clear summary of the article's findings; what the study adds to existing research and where and why the results may differ from previous research; strengths and limitations of the study; implications and next steps for research, clinical practice, and/or public policy; one-paragraph conclusion.

* l. 350, “In total, 72% of the participants had CKM.” – please note that this statement is repetitive of the prior paragraph.

* l.353: Please note that new results (e.g. Table S9) should not be presented in the Discussion. Please move them to the Results section, e.g. as a supplementary analysis.

* ll.362-364: Please see our earlier comment that this statement is not clear.

* l.373: Please note that you no longer need to cite relevant tables/figures in your discussion.

* l.388ff: Again, please note that new results (e.g. Table S10) should not be presented in the Discussion. Please move them to the Results section, e.g. as a supplementary analysis.

* l.392: Please avoid repeating numerical results in the Discussion.

* l.413ff: We feel that there is too much repetition of presenting the results rather than discussing the results.

* l.435: Why do you only mention the risk of ESKD as a strength and not discuss these results earlier in the discussion? Given that Taiwan has high dialysis prevalence and incidence rates, we think you could discuss these results earlier.

* l.460ff: Please see the comments above and include these findings in the results section.

General Editorial Requests

---

## [Editor Report · Decision Letter 4]

29 Apr 2025

Dear Dr. Wu,

Thank you very much for re-submitting your manuscript "Cardiovascular-Kidney-Metabolic Syndrome and Its Contribution to All-Cause and Cardiovascular Mortality: A Retrospective Cohort Study" (PMEDICINE-D-24-04247R4) for review by PLOS Medicine.

There are a few minor editorial issues that need to be addressed before we can accept the manuscript for publication; these are outlined at the end of this email. Please revise the paper accordingly, and submit the final revision until May 06.

Please ensure you address the specific points made by the editors. In your rebuttal letter you should indicate your response to the editors' comments and the changes you have made in the manuscript. Please submit a clean version of the paper as the main article file. A version with changes marked must also be uploaded as a marked up manuscript file. Please also check the guidelines for revised papers at http://journals.plos.org/plosmedicine/s/revising-your-manuscript for any that apply to your paper.

A reminder that when your manuscript is accepted, an uncorrected proof of your manuscript will be published online ahead of the final version, unless you've already opted out via the online submission form. If, for any reason, you do not want an earlier version of your manuscript published online or are unsure if you have already indicated as such, please let the journal staff know immediately at plosmedicine@plos.org.

If you have any questions in the meantime, please contact me directly at atosun@plos.org.

We look forward to receiving the revised manuscript.

Sincerely,

Alexandra Tosun, PhD

Associate Editor

PLOS Medicine

Requests from Editors:

1) Title: We suggest changing the title to: Cardiovascular-Kidney-Metabolic Syndrome and All-Cause and Cardiovascular Mortality: A Retrospective Cohort Study

2) Please ensure to update the funding statement in the online submission form (i.e. the metadata) with the statement provided in the rebuttal.

3) Please ensure to update the data availability statement in the online submission form (i.e. the metadata) with the statement provided in the rebuttal.

4) Thank you for providing your STROBE checklist. Please replace the page numbers with paragraph numbers per section (e.g. "Methods, paragraph 1"), since the page numbers of the final published paper may be different from the page numbers in the current manuscript.

5) Please remove any claims of novelty, e.g. on line 740 (“First, the study is a novel investigation…”).

6) The terms gender and sex are not interchangeable (as discussed in https://www.who.int/health-topics/gender#tab=tab_1 ); please use the appropriate term.

7) Please note that there are still instances of references to "men" and "women" instead of "male" and "female" instead of ". Please review and revise carefully (including tables and figures).

8) “Additionally, physical inactivity was more common among those with CKM syndrome (48.9%) compared to those without CKM (55.9%).” – Do you mean physical inactivity was more common among those without CKM syndrome?

9) “We assessed the distribution of the number of CKM components (hypertension, CKD, diabetes, metabolic syndrome, and hyperlipidemia) present in our cohort (Table S6).” – Without presenting the relevant denominators, it is impossible for the reader to understand how the results were calculated. Please revise the table accordingly. We think it would be more informative to simply present the results of how many people had at least 1, how many had 2, how many had 3, etc. of the components.

10) ll.308-312: “The increase in risk was even more pronounced for diabetes mortality, kidney-related mortality, and ESKD incidence. The aforementioned risk increased exponentially (Table 4 and Figure 2). The increase in risk was even more pronounced for diabetes mortality, kidney-related mortality, and ESKD incidence. The aforementioned risk increased exponentially (Figure 2).” - Please note that these are repetitive statements. Please revise.

11) l.356, “We compared the prevalence rates of CKM stages between the U.S. National Health and Nutrition Examination Survey (NHANES) [29] and our cohort…” – To guide the reader, we think it would be helpful to add a short sentence about why you compared these two cohorts in an additional analysis. This also applies to the following paragraphs.

---

## [Editor Report · Decision Letter 5]

6 May 2025

Dear Dr Wu, 

On behalf of my colleagues and the Academic Editor, David Flood, I am pleased to inform you that we have agreed to publish your manuscript "Cardiovascular-Kidney-Metabolic Syndrome and All-Cause and Cardiovascular Mortality: A Retrospective Cohort Study" (PMEDICINE-D-24-04247R5) in PLOS Medicine.

I appreciate your thorough responses to the reviewers' and editors' comments throughout the editorial process.

PRESS

Sincerely, 

Alexandra Tosun, PhD 

Associate Editor 

PLOS Medicine